# Industrial agglomeration and air pollution: A new perspective from enterprises in Atmospheric Pollution Transmission Channel Cities (APTCC) of Beijing-Tianjin-Hebei and its surrounding areas, China

Cuicui Xiao[1][⊗], Jingbo Zhou[iD][2][⊗]*, Xin Wang[3]*, Shumin Zhang[2]

1 School of Humanities and Social Sciences, University of Science and Technology Beijing, Beijing, China, 2 School of Environment and Natural Resources, Renmin University of China, Beijing, China, 3 China National Environmental Monitoring Centre, Beijing, China

⊗ These authors contributed equally to this work.
* zhoujb@ruc.edu.cn (JZ); wangxin@cnemc.cn (XW)

**Data Availability Statement:** The data used in this study are third-party data from third-party sources and can be accessed following the protocol

## Abstract

Air quality in China has gradually been improving in recent years; however, the Beijing-Tianjin-Hebei (BTH) region continues to be the most polluted area in China, with the worst air quality index. BTH and its surrounding areas experience high agglomeration of heavy-polluting manufacturers that generate electric power, process petroleum and coal, and carry out smelting and pressing of ferrous metals, raw chemical materials, chemical products, and non-metallic mineral products. This study presents evidence of the air pollution impacts of industrial agglomeration using the Ellison–Glaeser index, Herfindahl–Hirschman index, and spatial autocorrelation analysis. This was based on data from 73,353 enterprises in "2+26" atmospheric pollution transmission channel cities in BTH and its surrounding areas (herein referred to as BTH "2+26" cities). The results showed that Beijing, Yangquan, Puyang, Kaifeng, Taiyuan, and Jinan had the highest Ellison–Glaeser index among the BTH "2+26" cities; this represents the highest enterprise agglomeration. Beijing, Langfang, Tianjin, Baoding, and Tangshan also showed a low Herfindahl–Hirschman index of pollutant emissions, which have a relatively high degree of industrial agglomeration in BTH "2+26" cities. There was an inverted U-shaped relationship between enterprise agglomeration and air quality in the BTH "2+26" cities. This means that air quality improved with increased industrial agglomeration up to a certain level; beyond this point, the air quality begins to deteriorate with a decrease in industrial agglomeration.

## 1. Introduction

Industrial agglomeration promotes urban expansion, population concentration, and regional economic development; however, it also introduces a series of environmental problems [1–3]. As such, industrial agglomeration may have positive and negative environmental impacts. On

outlined in the Methods section (a) The urban economic and social development data (including total industrial output value, industrial structure, population density, and per capita GDP) are derived from the "China City Statistical Yearbook" issued by the National Bureau of Statistics in 2017, we can disclose this part in the attachment of supporting information. (b) the urban air quality data in the Beijing-Tianjin-Hebei region (including PM2.5, PM10, SO2, NO2) are collected from the National Urban Air Quality Real-time Release Platform of the Ministry of Ecology and Environment (http://106. 37.208.233:20035/), others can directly obtaining the data from http://106.37.208.233:20035/. (c) The individual enterprises data including the "Beijing-Tianjin-Hebei Air Pollution Transmission Channel Pollution Source Emission List" issued by China National Environmental Monitoring Centre (CNEMC), These kind of data cannot be shared publicly because it contains pollutant emission information of individual enterprises, which is sensitive and confidential, data are owned by China National Environmental Monitoring Centre (CNEMC), we have signed a confidentiality agreement with CNEMC to prohibit the disclosure of original data. others can contact the CNEMC to obtain the data (http://www.cnemc.cn/).

**Funding:** This study was supported by the Humanities and Social Sciences Project Youth Fund of Ministry of Education in China (18YJCZH196), the Fundamental Research Funds for the Central Universities (FRF-TP-19-069A1) "Application Research of Policy Analysis Methods and Techniques in Environmental Fields under the Background of Big Data" and Emission Characteristics and Total Amount Estimation of Non-key Industrial Enterprises (20170118).

**Competing interests:** The authors have declared that no competing interests exist.

the one hand, industrial agglomeration leads to greater regional resource consumption. It drives high density populations and the enlargement of the city scale which increase the consumption of raw materials, generating more pollution [4, 5]. On the other hand, it may improve the efficiency of resource utilization. Within industrial clusters, industrial agglomeration promotes the centralized use of raw materials, enhances economic scale, and creates knowledge spillover effects and technological advantages. Thus, it improves the overall efficiency of raw material and energy use in the region, reducing pollutant emissions [6, 7].

The industrial cluster theory first emerged in the mid-20th century, in which it proposed the concept of industrial externalities accompanied by geographic clusters [8–11]. Previous researchers have studied the effect of industrial aggregation; some recognize that industrial aggregation may promote the optimization of local industrial structures and improve economic and energy efficiency in agglomerated areas through economies of scale and knowledge spillover effects [12–16]. Specifically, industrial agglomeration promotes the scale effect of each industry in the region, introducing positive externalities, driving productivity growth and the energy efficiency of industries, and reducing local pollution [17, 18].

Other researchers observed the insignificant or negative effects of industrial agglomeration on energy efficiency [19, 20]. Researchers have reported that economic agglomeration and urbanization boosted energy demand and carbon emissions in more than 200 cities across the European Union (EU). An apparent positive correlation between industrial agglomeration and air pollution was observed; an increase in industrial agglomeration in 200 EU urban clusters was accompanied by serious air pollution. Researchers postulated that industrial agglomeration may increase the total amount of pollution within a certain geographic range. This would occur to the extent that it exceeds the local carrying capacity of the environment and generates negative externalities in terms of environmental pollution [21–23]. The scale of industrial production may inflate the discharge of industrial pollutants. The effect on industrial wastewater is considerably stronger than that on air pollution or waste [24].

Researchers have also observed that there is either a non-linear or unclear correlation between industrial agglomeration and environmental pollution [25–28]. A study found positive and negative relationships between agglomeration and energy efficiency from the paper and cement industries in Japan [29]. Another study suggested a non-linear relationship between agglomeration and productivity in Dutch firms [25]. It has been concluded that there is no definitive relationship between industrial agglomeration and pollution in the manufacturing industry [30]. A study examined the systematic linkage between industrial agglomeration and environmental performance using city-level data over the 2003–2010 period in China. They found a non-linear pattern between agglomeration and the emissions intensity of sulfur dioxide, and a "U-shaped" relationship between industrial agglomeration and environmental efficiency [2]. A study has also demonstrated that the relationship between emissions and manufacturing concentrations in the Beijing-Tianjin-Hebei (BTH) region is "inverted U-shaped," and this relationship is still in the preliminary stage [31]. While an increase in concentration will eventually deteriorate air quality, agglomeration may also promote the scale effect of the local area and greatly improve efficiencies in environmental technologies. In particular, an "inverted, U-shaped" relationship was observed between environmental pollution and income. This means that initially, environmental pollution worsened at lower income levels and then improved with economic growth exceeding a certain rate; this is known as the Environmental Kuznets Curve hypothesis [32].

In China, the BTH region experiences the worst air pollution in the country [33]. In 2019, 168 cities participated in an air quality index (AQI) evaluation; 16 of the 20 cities with the worst air quality were located in the BTH "2+26" cities, including Anyang, Xingtai, Shijiazhuang, Handan, Tangshan, Taiyuan, Zibo, Jiaozuo, Jincheng, Baoding, Jinan, Liaocheng,

Xinxiang, Hebi, Luoyang and Zhengzhou [34]. The average fine particulate matter ($PM_{2.5}$), coarse particulate matter ($PM_{10}$), sulfur dioxide ($SO_2$), and nitrogen dioxide ($NO_2$) concentrations in these 168 cities were 44, 74, and 33 μg/m³ respectively. The average $PM_{2.5}$, $PM_{10}$, $SO_2$, and $NO_2$ concentrations in the BTH "2+26" cities were 57, 100, 15, and 40 μg/m³. Highly polluting activities, such as steel making, iron making, and coking, alongside the cement, coal, and chemical industries are concentrated in the BTH and its surrounding areas [35]. The exposure pattern of the BTH cities showed that Beijing was prone to the highest air quality population exposure. Additionally, the exposure levels in Zhengzhou, Puyang, and Anyang were higher than the average of the BTH cities [36]. Due to variations in the contribution of unit emission reduction to pollutant concentration reduction between cities, there is heterogeneity in the marginal pollutant concentration reduction cost among various districts. A joint regional air pollution control policy in the BTH area is likely to save expenses associated with air pollution control compared with a locally based pollution control strategy [37].

In recent years, the government has been implementing the "*Beijing-Tianjin-Hebei (BTH) Collaborative Development Plan*" and the "*Opinions on Strengthening the Construction of the Key Platform for the Transfer of the Beijing-Tianjin-Hebei (BTH) Industry*;" prompting industrial agglomeration in BTH and its surrounding areas. A comprehensive mechanism to prevent and control air pollution in the BTH region and its surrounding areas is crucial to improve air quality [38, 39]. The "*Rules for the Implementation of the Action Plan for the Prevention and Control of Air Pollution in BTH and Surrounding Areas*" proposes that BTH and its surrounding areas should significantly reduce the total emissions from $SO_2$, nitrogen oxides ($NO_x$), PM, and volatile organic compounds (VOCs) [40]. The BTH region is a key area for air pollution prevention and control as it consumes over 33% of coal in China. Additionally, the main air pollutant emissions account for approximately 30% of the total emissions in China, and the emissions intensity per square kilometer is approximately four times the national average [41]. Rapid urbanization with high energy consumption has imposed significant pressure on ecosystems [42].

Although there has been substantial progress on collaborative governance policies in the BTH region, the comprehensive reduction of regional pollutant emissions remains the focus and the greatest challenge. The "*Enhanced Measures for the Prevention and Control of Air Pollution in Beijing-Tianjin-Hebei (BTH) (2016–2017)*" and the "*Work Plan for Air Pollution Control in Beijing-Tianjin-Hebei (BTH) and Surrounding Areas in 2017*," emphasize the air pollution control task for the BTH and its surrounding areas. With coordinated development of various industries, the agglomeration effect in the BTH has gained prominence over recent years. Whether industrial agglomeration has intensified air pollution in the BTH "2+26" cities is a topic worthy of in-depth study.

It is important to understand the overall impact of industrial agglomeration on the BTH regional environment, and whether it exacerbates air quality. To clarify this relationship, we employed data on 73,353 enterprises in the BTH region of China and compared the spatial emission characteristics of industrial pollution sources by using the Ellison–Glaeser index (EGI), Herfindahl–Hirschman index (HHI), and spatial autocorrelation analysis. We then constructed a spatial measurement model that empirically analyzed the relationship between industrial agglomeration and pollution aggregation. The effect of industrial aggregation on air quality was evaluated, providing a new perspective on regional atmospheric environmental governance.

This study differs from existing studies on agglomeration and environmental performance in two key ways. First, at the data level, previous studies on industrial agglomeration largely utilized city-level panel data; this study used the emissions data of 73,353 enterprises in BTH "2+26" cities. Then, we analyzed the spatial distribution of the agglomeration and its pollutant

emission accumulation in different areas. Second, we constructed a spatial measurement model on the impact of industrial agglomeration on environmental pollution. This enabled further exploration into the impact of enterprise agglomeration, emission agglomeration, and air quality, characterizing the degree of pollution agglomeration in different areas. It aims to provide policy support for the coordinated management of air pollution in BTH and its surrounding cities in China.

## 2. Materials and methods

### 2.1 Study areas

The "*Air Pollution Prevention and Control Work Plan of Beijing-Tianjin-Hebei and Surrounding Areas in 2017*," defines the scope of remediation in the BTH atmospheric pollution transmission channel cities (APTCC), which includes Beijing, Tianjin, Shijiazhuang, Tangshan, Langfang, Baoding, Cangzhou, Hengshui, Xingtai, Handan, Taiyuan, Yangquan, Changzhi, Jincheng, Jinan, Zibo, Jining, Dezhou, Liaocheng, Binzhou, Heze, Zhengzhou, Kaifeng, Anyang, Hebi, Xinxiang, Jiaozuo, and Puyang (i.e., the BTH "2+26" C cities) (Fig 1). There are 160 urban air quality stations in the BTH "2+26" cities, as shown in Fig 1.

### 2.2 Data sources and description

The data used in this study included industrial enterprise pollutant emission data, urban economic and social development data, and air quality data from the BTH "2+26" cities in 2016.

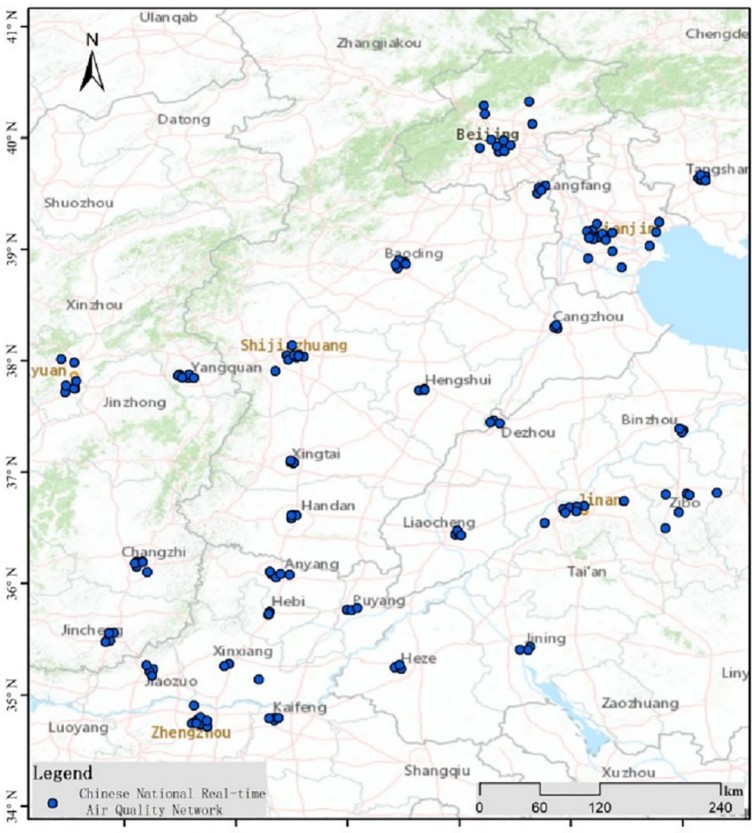

**Fig 1. Locations of air quality monitoring sites in the BTH "2+26" cities.** The map was prepared in ARCGIS using political boundaries from the Global Database of Administrative Areas (https://resources.arcgis.com/en/help/).

**Table 1. Basic information on the "2+26" cities of Beijing-Tianjin-Hebei and its surrounding areas.**

| Province | City | Latitude (°N) | Longitude (°E) | Urban area (km²) | Gross industrial output value / 100 million yuan |
|---|---|---|---|---|---|
| Beijing | Beijing | 40.03 | 116.38 | 16,410 | 18,087.3 |
| Tianjin | Tianjin | 39.08 | 117.38 | 11,760 | 27,401.7 |
| Hebei | Baoding | 38.87 | 115.47 | 2,565 | 4,721.5 |
| | Cangzhou | 38.31 | 116.87 | 183 | 5,690.89 |
| | Handan | 36.61 | 114.51 | 2,648 | 5,055.07 |
| | Hengshui | 37.74 | 115.67 | 1,510 | 1,793.29 |
| | Langfang | 39.54 | 116.74 | 960 | 3,791.87 |
| | Shijiazhuang | 38.03 | 114.5 | 2,194 | 9,644.79 |
| | Tangshan | 39.64 | 118.18 | 3,874 | 9,967.68 |
| | Xingtai | 37.09 | 114.5 | 132 | 2,933.65 |
| Henan | Anyang | 36.08 | 114.37 | 544 | 3,991.03 |
| | Hebi | 35.73 | 114.3 | 679 | 2,004.21 |
| | Jiaozuo | 35.22 | 113.23 | 544 | 5,893.24 |
| | Kaifeng | 34.79 | 114.34 | 1,837 | 3,013.27 |
| | Puyang | 35.77 | 115.05 | 263 | 3,818.17 |
| | Xinxiang | 23.47 | 76.03 | 346 | 4,535.11 |
| | Zhengzhou | 34.77 | 113.67 | 1,010 | 14,465.6 |
| Shandong | Binzhou | 37.38 | 117.99 | 3,763 | 7,382.03 |
| | Dezhou | 37.46 | 116.33 | 1,752 | 10,569.5 |
| | Heze | 35.25 | 115.46 | 2,261 | 7,934.16 |
| | Jinan | 36.66 | 116.99 | 5,112 | 5,486.56 |
| | Jining | 35.42 | 116.6 | 1,648 | 5,499.65 |
| | Liaocheng | 36.45 | 115.99 | 1,710 | 8,999.91 |
| | Zibo | 36.73 | 118.01 | 2,989 | 12,011.5 |
| Shanxi | Jincheng | 35.51 | 112.86 | 2,164 | 841.87 |
| | Taiyuan | 37.86 | 112.5 | 1,500 | 2,207.42 |
| | Yangquan | 37.86 | 113.54 | 652 | 536.72 |
| | Changzhi | 36.17 | 113.14 | 334 | 1,543.79 |

Data Source: "The 2016 Urban Statistical Yearbook of the People's Republic of China" issued by the National Bureau of Statistics in 2017.

The enterprise and pollution source emission inventory datasets were derived from the China National Environmental Monitoring Centre (due to data confidentiality, this research data has not been disclosed in this study). Note that to obtain this data, the consent of the China National Environmental Monitoring Centre is required. Urban economic and social development data (including total industrial output values, industrial structure, population density, and per capita gross domestic product (GDP)) were collated from the National Bureau of Statistics from the Urban Statistical Yearbook of China. Air quality data (including $PM_{2.5}$, $PM_{10}$, $SO_2$, $NO_2$, ozone ($O_3$), and carbon monoxide (CO)) in the BTH "2+26" cities were collected from the National Urban Air Quality Real-time Publishing Platform of the Ministry of Ecology and Environment (http://106.37.208.233:20035/). Table 1 provides basic information on the "2 +26" cities of the BTH and its surrounding areas.

## 2.3 Methods and models

**2.3.1 Herfindahl index.** The HHI was used measure the agglomeration of pollutant emissions of BTH "2+26" cities, and explore the different degrees of agglomeration between key and non-key survey micro-enterprises. Compared with the EGI, the HHI is simple to calculate and

can more accurately reflect changes in industry or corporate market concentrations. HHI was selected to measure the agglomeration of pollutant emissions. This was because we had obtained the pollutant emission data of 73,353 micro-enterprises, and HHI is better able to measure the pollution agglomeration of micro-enterprises in the region. The HHI was calculated using Eq (1):

$$H = \sum_{i}^{N} S_i^2 = \sum_{i=1}^{N} \left(\frac{X_i}{X}\right)^2 \tag{1}$$

Here, $X_i$ is the pollutant emission of enterprise, $i$, in a specific industry; $X$ is the total pollutant emission of enterprise $I$; $S_i$ is the proportion of a specific pollutant discharge of enterprise $I$; and $N$ is the total number of enterprises in this industry. The range of HHI is $0<H\leq1$; that is, when $H$ is equal to 1, pollutant emissions are highly concentrated within a specific city. When $H$ is close to 0, the scale of enterprises in a specific city is similar. The study calculated the HHI for key and non-key surveyed enterprises separately, as enterprises with large emissions may have an extreme impact on HHI. In addition, pollutant emission levels in different industries vary; as such, the HHI in different industries was also calculated.

**2.3.2 Ellison and Glaeser index.** Based on the EG coefficients proposed by Ellison and Glaeser in 1997 [43], an adjusted EG was used to calculate the degree of enterprise concentration in each city [44]. The minimum unit of calculation was at the county (district) level. Assuming that there are $N$ enterprises in industry, $i$, and there are $r$ counties (districts) in an administrative division, the EG coefficient of industry, $i$, was calculated using Eq (2):

$$EG = \frac{G_i - [1 - \sum_{j=1}^{r} X_i^2]H}{[1 - \sum_{j=1}^{r} X_i^2](1 - H)} \tag{2}$$

where $G_i$ is calculated using Eq (3), and H is calculated using Eq (4):

$$G_i = \sum_{j=1}^{r} (X_j - S_{ij})^2 \tag{3}$$

$$H = \sum_{k=1}^{N} Z_k^2 \tag{4}$$

Here, $G$ is the Gini coefficient; $H$ is the HHI; $X_j$ is the ratio of the output value of the district (county) of $j$ to the total output value of all districts and counties of the city in that year; $S_{ij}$ is the ratio of the output value of industry, $i$, to the total output value of the industry in the district (county); $Z_k$ is the ratio of the output value of enterprise, $k$, to the total output value of industry, $i$. We assume that all enterprises in industry, $i$, have the same scale; this means their industrial output values are equal. This assumption was made in the formula to obtain HHI as detailed data on state-owned and non-state-owned industrial enterprises above a designated size are not publicized in China [45]. The adjusted formula for the HHI was calculated using Eq (5):

$$H = \sum_{j=1}^{r} n_{ij} \left(\frac{Output_{ij}/n_{ij}}{Output_i}\right)^2 \equiv \sum_{j=1}^{r} \frac{1}{n_{ij}} \left(\frac{Output_{ij}}{Output_i}\right)^2 \equiv \sum_{j=1}^{r} \frac{1}{n_{ij}} S_{ij}^2 \tag{5}$$

Here, $i$ represents an industry; $j$ represents the district or county; $r$ represents the total districts or counties in the city; $n_{ij}$ is the number of enterprises in county, $j$; $Output_{ij}$ is the total output value of the second industry of industry, $i$, in county $j$; $Output_i$ is the total output value of the second industry of industry, $i$, in the city. HHI may reflect changes in the industry or enterprise market concentration, market monopoly, and scale of competition.

**2.3.3 Spatial correlation test.** We used global and local autocorrelation methods to analyze if there was spatial accumulation of "high pollution around high-pollution areas." This was carried out to verify the spatial correlation of pollutant emissions among different regions

in the "2+26" cities. Typically, global autocorrelation is used to reflect the presence of research problems within the entire research scope. The Moran's I index and its numerical value were calculated to determine whether the research object has a spatial correlation. The Moran's I index was selected to analyze the spatial correlation of different pollutants in the "2+26" cities. The Global Moran's I was calculated using Eq (6):

$$Global\ Moran's\ I = \frac{\sum_{i=1}^{n}\sum_{j=1}^{n}W_{ij}(Y_i - \bar{Y})(Y_j - \bar{Y})}{S^2\sum_{i=1}^{n}\sum_{j=1}^{n}W_{ij}} \tag{6}$$

The Local Moran's I was calculated using Eq (7):

$$Local\ Moran's\ I_i = \frac{(Y_i - \bar{Y})}{S^2}\sum_{j=1}^{n}W_{ij}(Y_j - \bar{Y}) \tag{7}$$

The $S^2$ and $\bar{Y}$ values were calculated using Eqs (8) and (9), respectively.

$$S^2 = \frac{1}{n}\sum_{i=1}^{n}(Y_i - \bar{Y})^2 \tag{8}$$

$$\bar{Y} = \frac{1}{n}\sum_{i=1}^{n}Y_i \tag{9}$$

Here, $Y_i$ represents the pollutant concentration of city, $i$; $n$ is the total number of cities; and $W_{ij}$ is the spatial weight matrix. When $W_{ij} = 1$, this indicates that the two areas are adjacent. If $W_{ij} = 0$, it indicates that the two areas are not adjacent or that $i$ and $j$ are the same area. $Y_i$ and $Y_j$ represent specific observations on enterprises in the BTH "2+26" cities.

The value range of Moran's I is (-1, 1); when the value of Moran's I is greater than zero, observations between the enterprises of the region have a positive correlation, and this spatial correlation strengthens as the value approaches 1. When the value is less than zero, observations between the regions have a negative correlation, and this negative correlation strengthens as the value tends to -1. When the value is zero, there is no spatial correlation between different cities.

For Local Moran's I, if the Local Moran's I >0, it indicates that there is a positive spatial effect. Cities with similar pollutant concentrations are aggregated together, showing high-high aggregation or low-low aggregation. If the Local Moran's I <0, it indicates that the spatial effect is negative, whereby different pollutant concentrations are clustered together; this shows high-low aggregation and low-high aggregation.

To accurately determine spatial correlation, the Moran's I index construction statistic was used to conduct the significance test. The expectations, $E(I)$, and variances, $VAR(I)$, of Moran's I were calculated using Eqs (10)–(15):

$$E(I) = -\frac{1}{(n-1)} \tag{10}$$

$$VAR(I) = \frac{n^2 w_1 + n w_2 + 3 w_0^2}{w_0^2(n^2 - 1)} - E^2(I) \tag{11}$$

$$Here,\ w_o = \sum_{i=1}^{n}\sum_{j=1}^{n}W_{ij} \tag{12}$$

$$w_1 = \frac{1}{2}\sum_{i=1}^{n}\sum_{j=1}^{n}(W_{ij}+W_{ji})^2 \tag{13}$$

$$w_2 = \sum_{k=1}^{n}(W_{ik}+W_{ki})^2 \tag{14}$$

$$Z = \frac{I - E(I)}{\sqrt{VAR(I)}} \tag{15}$$

In Eqs (10)–(15), the *E(I)* and *VAR(I)* are the expectations and variances of Moran's I; $n$ is the total number of cities; $W_{ij}$, $W_{ji}$, $W_{ik}$, and $W_{ki}$ are the spatial weight matrices. We constructed a $Z$ statistic that progressively obeyed a normal distribution. A value of the $Z$ statistic greater than the critical value of the normal distribution at the 0.05 confidence level indicates that the evaluation index has spatial autocorrelation. The local autocorrelation test calculated the spatial agglomeration effect by calculating the local Moran's I using the Moran scattergram.

**2.3.4 Spatial measurement model analysis.** The spatial lag model (SLM) was used to investigate whether the variable has spread (spillover effect) in the area, and its expression is given by Eq (16):

$$Y = \rho WY + X\beta + \varepsilon \tag{16}$$

Here, $Y$ is the dependent variable (air pollution); $X$ is the exogenous explanatory variable matrix of order $n \times k$; and $\rho$ is the spatial lag coefficient, which reflects the spatial dependence degree of the agglomeration between BTH "2+26" cities. It is the direction and extent of influence from the observation value, $WY$, in the adjacent area on the observation value, $Y$, in the specific area. $W$ is a spatial weight matrix of order $n \times n$, $WY$ is the spatial lagging dependent variable, *and* $\varepsilon$ is the random error term vector.

The mathematical expressions of the spatial error model (SEM) are shown in Eqs (17) and (18):

$$Y = X\beta + \varepsilon \tag{17}$$

$$\varepsilon = \lambda W\varepsilon + \mu \tag{18}$$

Here, $\beta$ is the influence of the explanatory variable, $X$, on the explained variable, $Y$; $\varepsilon$ is the random error term vector; and $\lambda$ is the spatial error coefficient, which measures the spatial dependence of BTH "2+26" cities. This means the influence of the observed value, $Y$, in neighboring regions on the observed value, $Y$, in the specific region.

Based on the quadratic items of enterprise agglomeration, we established the following basic spatial lag models (SLM) given in Eqs (19) and (20):

$$Y_i = \beta_0 + \rho WY_i + \beta_1 Aggregation + \beta_2 Aggregation^2 + \beta_3 Z_i + \varepsilon_i \tag{19}$$

$$Y_i = \beta_0 + \rho WY_i + \beta_1 Aggregation + \beta_2 Z_i + \varepsilon_i \tag{20}$$

Here, *Aggregation* represents the enterprise agglomeration index and the enterprise pollution concentration agglomeration index; and $Z_i$ represents the control variable, which is per capita GDP (perGDP), population density (POP), and industrial structure (IND). We used EG

**Table 2. Variable definition and explanation.**

|  | Definition of Variables | Unit |
|---|---|---|
| Dependent variables | Annual average of air quality index (AQI) | not applicable (n/a) |
| Explanatory variables | Enterprise agglomeration EG degree (EG) | not applicable (n/a) |
|  | Per capita GDP (perGDP) | RMB Yuan |
|  | Population density (POP) | Person/km$^2$ |
|  | Ratio of the gross domestic product of the secondary industry to GDP (IND) | % |

to represent the enterprise agglomeration index, and HHI to represent the enterprise pollution concentration agglomeration index (Table 2).

## 3. Results and discussion

### 3.1 Industrial enterprise clustering index

The EG index provides a well-defined range for measuring the geographical distribution of industrial agglomerations. According to Ellison and Glaeser [43], the EG index may be divided into three categories: (1) a low agglomeration is when the EG index is less than 0.02; (2) a moderate agglomeration is when the EG index exceeds 0.02 and is less than 0.05; and (3) a relatively high agglomeration is when the EGI is greater than 0.05.

Based on the adjusted EG index calculation, we used county (district) level data to analyze the agglomeration of industrial enterprises in BTH "2+26" cities in 2016; there are 303 districts and counties in the BTH "2+26" cities. The results show that there were 20 cities with an EG index below 0.02, four cities between 0.02 and 0.05, and four cities with an EG index exceeding 0.05 (Fig 2). The EG index of Beijing, Yangquan, Puyang, and Kaifeng had the highest concentrations among the "2+26" cities.

In general, the agglomeration of manufacturing enterprises in the BTH region was relatively low (Table 3). Among the 28 cities, 20 had a low EGI of less than 0.02, indicating that industrial agglomeration in this region was relatively low. According to the industry categories of enterprises, more than 85% of the BTH "2+26" cities were manufacturing enterprises (65,275 of 73,353 enterprises) (Table 4); however, BTH manufacturing enterprises are not highly agglomerated. The proportion of tertiary industry in the economic development of the BTH area was relatively high, consistent with the overall trend of transformation from secondary to tertiary industries all over the country. The threshold for entry into labor-intensive industries was relatively low [46]. Beijing was faster and better than other areas in terms of industrial structure transformation in the BTH area [47]. The employment and industrial structure in Hebei province was not coordinated, characterized by a prominent labor and resource-intensive nature.

The results show that there are two major types of cities with a high agglomeration of enterprises. The first category includes cities in the Shanxi, Henan, and Shandong provinces; these enterprises form economies of scale and a certain level of agglomeration. Although these regions have relatively higher industrial agglomeration, they are mostly labor and resource-intensive [48]. They were mostly in the second half of the medium term of industrialization or the first half of the late term of industrialization, with a relatively lagging tertiary industry and lower innovation output [49, 50]. The other category includes the areas surrounding Beijing and Tianjin. Due to adjustments in the urban and industrial planning policies, enterprises in the core areas of Beijing have relocated to its peripheries. A considerable number of small and

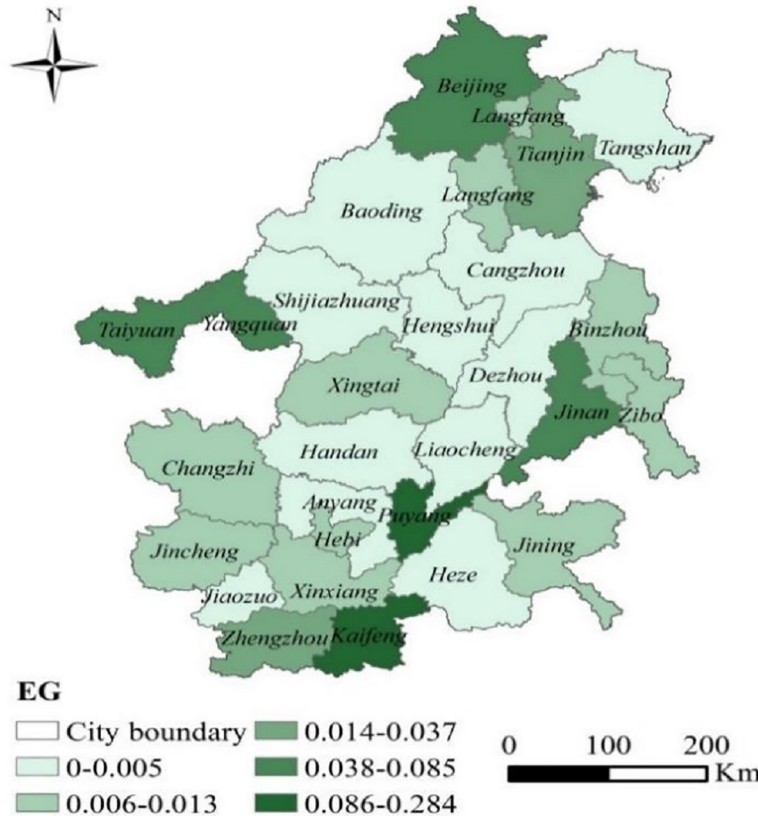

**Fig 2. EG index of the BTH "2+26" cities.** The map was prepared in ARCGIS using political boundaries from the Global Database of Administrative Areas (https://resources.arcgis.com/en/help/).

medium-sized enterprises have gathered in the suburbs of Beijing. The manufacturing industry plays a key role in the industrial shift in the BTH region, decentralizing the non-capital functions of Beijing to other areas such as Tianjin and Hebei. Hebei, Shandong, and Henan have all become destinations for this type of industrial shift from Beijing [51–54].

## 3.2 Spatial density analysis of air pollutant emissions

We divided the enterprise statistics into industrial sectors according to the "National Economic Industry Classification GBT 4754–2017." The surveyed enterprises were from 44 industries (Table 4); among them, manufacturing enterprises accounted for more than 85%, covering non-metallic mineral products, metal products, general equipment manufacturing, and rubber and plastic products. There were 6,604 electricity, heat production and supply enterprises in the BTH "2+26" cities (Table 4), releasing the heaviest $SO_2$ and $NO_x$ emissions.

We obtained the pollution discharge list of the 73,353 industrial enterprises in the BTH "2+26" cities for 2016 and statistics on the pollutant emissions from different industries. The results of the analysis are presented in Table 5.

In terms of emission types, electricity, heat production and supply, ferrous metal smelting and rolling processing, and non-metallic mineral products were the major sources of $SO_2$, $NO_x$, and $PM_{2.5}$ emissions among the "2+26" cities. The $SO_2$ emissions from the electricity, heat production, and supply industries were 315,136.7 t, accounting for 31.8% of $SO_2$ emissions in the industry. The $NO_x$ emissions were 522,941.1 t, accounting for 31.8% of the overall industrial $NO_x$ emissions (Table 5).

**Table 3. EG Index in BTH and the surrounding "2+26" cities.**

| EG range | City | Ellison–Glaeser Index (EGI) | Agglomeration classification |
|---|---|---|---|
| EG<0.02 | Dezhou | 0.000 | Low agglomeration |
| | Heze | 0.000 | |
| | Baoding | 0.002 | |
| | Shijiazhuang | 0.002 | |
| | Liaocheng | 0.003 | |
| | Cangzhou | 0.003 | |
| | Jiaozuo | 0.003 | |
| | Handan | 0.003 | |
| | Hengshui | 0.003 | |
| | Tangshan | 0.005 | |
| | Anyang | 0.005 | |
| | Zibo | 0.006 | |
| | Xinxiang | 0.006 | |
| | Xingtai | 0.007 | |
| | Hebi | 0.007 | |
| | Langfang | 0.008 | |
| | Binzhou | 0.009 | |
| | Jining | 0.010 | |
| | Changzhi | 0.013 | |
| | Jjincheng | 0.013 | |
| 0.02≤EG≤0.05 | Tianjin | 0.024 | Moderate agglomeration |
| | Zhengzhou | 0.037 | |
| | Jinan | 0.047 | |
| | Taiyuan | 0.049 | |
| EG>0.05 | Beijing | 0.054 | High agglomeration |
| | Yangquan | 0.063 | |
| | Puyang | 0.085 | |
| | Kaifeng | 0.284 | |

The results showed that $SO_2$ and $NO_x$ emissions from the power industry accounted for the largest proportion of total emissions in 2016. Although the government has adopted various measures to promote the de-capacity of steel, cement, and petrochemical plants in BTH during recent years, these industries continue to cause the highest $SO_2$ and $NO_x$ emissions and are the main sources of air pollution in the BTH [55]. Desulfurization and denitrification facilities have been installed in the coal-fired power plants in China, with 80% of plants completing ultra-low emission retrofits by 2019 [56–58]. The $PM_{2.5}$ emissions of ferrous metal smelting and rolling processing industries were 269,447 t, accounting for 31.6% of the overall industrial $PM_{2.5}$ emissions, and causing the highest $PM_{2.5}$ pollution (Table 5). Emissions from steel companies have a significant impact on urban air quality [59]; the air quality of the 20 cities with the highest production capacity in the steel industry (where capacity accounts for 51% of the total production capacity in China), exceeded the air quality standards of China [60]. Therefore, it is necessary to accelerate the ultra-low emission transformation in the steel industry to win the Blue-Sky Protection Campaign [61, 62] and implement coordinated emission reduction policies for this industry in the BTH region [63].

### 3.3 Agglomeration index of pollutant emissions

We used the HHI to measure the agglomeration of pollutant emissions in different cities. Based on the classification standard of the Ministry of Ecology and Environment for air

**Table 4. Distribution of industrial enterprises in the BTH "2+26" cities.**

| Industry | Number of enterprises | Proportion (%) |
|---|---|---|
| Manufacturing | **65,275** | **88.19** |
| Non-metallic mineral products | 9,287 | 12.55 |
| Metal products | 6,279 | 8.48 |
| General equipment manufacturing | 5,254 | 7.10 |
| Rubber and plastic products | 5,247 | 7.09 |
| Manufacturing of chemical materials and products | 4,489 | 6.06 |
| Wood processing and wood, bamboo, rattan, brown, grass products | 4,032 | 5.45 |
| Other manufacturing | 3,408 | 4.60 |
| Special equipment manufacturing | 3,060 | 4.13 |
| Agricultural and sideline food processing | 2,984 | 4.03 |
| Food manufacturing | 2,293 | 3.10 |
| Electrical machinery and equipment manufacturing | 2,188 | 2.96 |
| Ferrous metal smelting and rolling processing | 2,173 | 2.94 |
| Textile | 2,046 | 2.76 |
| Furniture manufacturing | 1,552 | 2.10 |
| Automobile manufacturing | 1,548 | 2.09 |
| Reproduction of printing and recording media | 1,464 | 1.98 |
| Papermaking and paper products | 1,316 | 1.78 |
| Textile clothing, clothing | 1,043 | 1.41 |
| Metalwork, machinery and equipment repair | 889 | 1.20 |
| Leather, fur, feather and other products and footwear | 785 | 1.06 |
| Pharmaceutical manufacturing | 780 | 1.05 |
| Culture and education, industrial beauty, sports and entertainment goods manufacturing | 635 | 0.86 |
| Non-ferrous metal smelting and rolling processing | 630 | 0.85 |
| Wine, beverage and refined tea manufacturing | 589 | 0.80 |
| Oil, coal and other fuel processing | 400 | 0.54 |
| Manufacturing of railways, ships, aerospace and other transport equipment | 237 | 0.32 |
| Manufacturing of computers, communications and other electronic equipment | 222 | 0.30 |
| Chemical fiber manufacturing | 178 | 0.24 |
| Instrumentation manufacturing | 142 | 0.19 |
| Comprehensive utilization of waste resources | 117 | 0.16 |
| Tobacco products | 8 | 0.01 |
| Electricity, heat, gas and water production and supply | **6,634** | **8.96** |
| Electricity and heat production and supply | 6,604 | 8.92 |
| Gas production and supply | 30 | 0.04 |
| Mining | **1,444** | **1.95** |
| Coal mining and washing | 1,068 | 1.44 |
| Non-metallic mining | 157 | 0.21 |
| Other mining | 141 | 0.19 |
| Ferrous metal mining | 45 | 0.06 |
| Non-ferrous metal mining | 18 | 0.02 |
| Mining specialty and auxiliary activities | 15 | 0.02 |
| Other | **664** | **0.90** |
| Total | **74,017** | **100.00** |

Data sources: The enterprise and the pollution source emission inventory data were derived from the environmental statistical report of the Ministry of Ecology and Environment in 2017.

**Table 5. Pollutant emissions from different industries of BTH "2+26" cities in 2016.**

| Industry | $SO_2$ emissions (t) | Percentage of $SO_2$ emissions in different industries (%) | $NO_x$ emissions (t) | Percentage of $NO_x$ emissions in different industries (%) | $PM_{2.5}$ emissions (t) | Percentage of $PM_{2.5}$ emissions in different industries (%) | CO emissions (t) | Percentage of CO emissions in different industries (%) | VOC emissions (t) | Percentage of VOC emissions in different industries (%) |
|---|---|---|---|---|---|---|---|---|---|---|
| Electricity, heat production and supply | 315,136.7 | 31.78 | 522,941.1 | 32.84 | 122,151.6 | 14.31 | 771,574.5 | 4.92 | 37,563.9 | 2.51 |
| Ferrous metal smelting and rolling processing | 261,251.9 | 26.34 | 311,288.1 | 19.55 | 269,447 | 31.56 | 10,719,702 | 68.39 | 203,135.4 | 13.59 |
| Non-metallic mineral products | 136,932.4 | 13.81 | 230,454.1 | 14.47 | 155,959.6 | 18.27 | 2,283,086 | 14.57 | 226,871.1 | 15.18 |
| Oil, coal, and other fuel processing | 61,608.52 | 6.21 | 131,603.4 | 8.27 | 46,188.34 | 5.41 | 217,514.6 | 1.39 | 215,045.7 | 14.39 |
| Chemical raw materials and chemical manufacturing | 48,745.17 | 4.92 | 101,931.2 | 6.40 | 37,782.47 | 4.43 | 196,108.7 | 1.25 | 300,512.7 | 20.10 |
| Non-ferrous metal smelting and rolling processing | 28,215.59 | 2.85 | 47,752.47 | 3.00 | 55,677.07 | 6.52 | 483,159.3 | 3.08 | | |

Data sources: The enterprise and the pollution source emission inventory data were derived from the environmental statistical report of the Ministry of Ecology and Environment in 2017.

pollution control enterprises [64], a total of 73,353 industrial enterprises in BTH were categorized as 11,168 key surveyed enterprises and 62,185 non-key surveyed enterprises. The HHI was used to calculate the industrial pollution discharge agglomeration of the "2+26" cities; the results are presented in Table 6.

The results showed that the overall concentration of pollutant emissions in the BTH "2+26" cities was high. In terms of different pollutants, the overall agglomeration in northwest BTH was low, while the central and southeastern parts were high; there were also specific differences. The overall agglomeration of key and non-key survey enterprises was similar, although identical trends were not observed for various cities (Fig 3).

Beijing, Langfang, Tianjin, Baoding, and Tangshan had a low HHI of pollutant emissions, which indicates a relatively high degree of pollution agglomeration. This is closely related to the AQI in these cities [65, 66], as the large number of pollutant emission sources causes poor air quality [40]. The HHI of key industries in Tangshan, Zibo, Tianjin, Baoding, Changzhi, Xingtai, Beijing, Handan, and Shijiazhuang showed higher pollution concentration aggregation than other cities. As there were a large number of non-key industries, the HHI of these industries was determined separately. It was observed that Beijing, Tianjin, Langfang, Baoding, Hengshui, Xingtai, and Zhengzhou had high levels of pollution agglomeration; there was a correlation between industrial clusters and pollution clusters [2, 3, 7]. Therefore, follow-up studies analyzed and verified the correlation between industrial agglomeration and pollution agglomeration.

**Table 6. Agglomeration of pollutant emissions.**

| City | $H_0$ | $SO_2$-$H_0$ | $NO_X$-$H_0$ | $PM_{2.5}$-$H_0$ | $H_1$ | $SO_2$-$H_1$ | $NO_X$-$H_1$ | $PM_{2.5}$-$H_1$ |
|---|---|---|---|---|---|---|---|---|
| Beijing | 0.07 | 0.07 | 0.08 | 0.07 | 0.01 | 0.01 | 0.01 | 0.01 |
| Tangshan | 0.03 | 0.03 | 0.03 | 0.02 | 0.09 | 0.12 | 0.09 | 0.07 |
| Langfang | 0.17 | 0.14 | 0.14 | 0.23 | 0.02 | 0.01 | 0.02 | 0.02 |
| Tianjin | 0.06 | 0.05 | 0.04 | 0.09 | 0.02 | 0.02 | 0.02 | 0.02 |
| Baoding | 0.06 | 0.05 | 0.05 | 0.07 | 0.04 | 0.02 | 0.09 | 0.01 |
| Cangzhou | 0.19 | 0.09 | 0.26 | 0.21 | 0.09 | 0.08 | 0.18 | 0.02 |
| Shijiazhuang | 0.08 | 0.08 | 0.07 | 0.10 | 0.18 | 0.08 | 0.40 | 0.07 |
| Yangquan | 0.21 | 0.24 | 0.20 | 0.19 | 0.13 | 0.16 | 0.17 | 0.06 |
| Taiyuan | 0.17 | 0.19 | 0.08 | 0.25 | 0.62 | 0.40 | 0.71 | 0.75 |
| Hengshui | 0.10 | 0.06 | 0.10 | 0.13 | 0.05 | 0.05 | 0.08 | 0.02 |
| Binzhou | 0.09 | 0.05 | 0.05 | 0.16 | 0.07 | 0.05 | 0.10 | 0.05 |
| Dezhou | 0.14 | 0.06 | 0.18 | 0.17 | 0.29 | 0.35 | 0.39 | 0.12 |
| Xingtai | 0.12 | 0.04 | 0.05 | 0.27 | 0.05 | 0.03 | 0.09 | 0.03 |
| Jinan | 0.18 | 0.13 | 0.25 | 0.16 | 0.09 | 0.02 | 0.09 | 0.16 |
| Zibo | 0.05 | 0.04 | 0.03 | 0.07 | 0.15 | 0.21 | 0.19 | 0.07 |
| Liaocheng | 0.09 | 0.07 | 0.08 | 0.12 | 0.13 | 0.13 | 0.20 | 0.05 |
| Handan | 0.07 | 0.08 | 0.08 | 0.06 | 0.10 | 0.17 | 0.11 | 0.02 |
| Changzhi | 0.06 | 0.07 | 0.04 | 0.06 | 0.11 | 0.12 | 0.11 | 0.11 |
| Anyang | 0.16 | 0.16 | 0.12 | 0.19 | 0.22 | 0.27 | 0.24 | 0.14 |
| Hebi | 0.10 | 0.09 | 0.11 | 0.11 | 0.13 | 0.04 | 0.30 | 0.04 |
| Puyang | 0.30 | 0.38 | 0.26 | 0.26 | 0.20 | 0.10 | 0.18 | 0.33 |
| Jincheng | 0.11 | 0.16 | 0.10 | 0.07 | 0.21 | 0.27 | 0.20 | 0.17 |
| Jining | 0.10 | 0.09 | 0.10 | 0.12 | 0.45 | 0.39 | 0.39 | 0.58 |
| Heze | 0.08 | 0.06 | 0.10 | 0.07 | 0.22 | 0.18 | 0.45 | 0.03 |
| Xinxiang | 0.06 | 0.07 | 0.05 | 0.05 | 0.14 | 0.17 | 0.18 | 0.06 |
| Jiaozuo | 0.10 | 0.12 | 0.08 | 0.10 | 0.08 | 0.11 | 0.10 | 0.04 |
| Zhengzhou | 0.14 | 0.15 | 0.13 | 0.13 | 0.05 | 0.05 | 0.05 | 0.07 |
| Kaifeng | 0.31 | 0.35 | 0.38 | 0.22 | 0.50 | 0.56 | 0.87 | 0.09 |

Notes: $H_0$ denotes the overall agglomeration of key enterprises, and $H_1$ denotes the overall agglomeration of non-key survey enterprises.

### 3.4 Spatial correlation of pollutant emissions of industrial enterprises

**3.4.1 Spatial autocorrelation test.** The pollutant emissions of the BTH "2+26" cities demonstrated a certain level of spatial agglomeration. To further verify this spatial correlation, Moran's I was used to confirm the relationship between district and county-level pollutant emissions and spatial distribution in the BTH "2+26" cities. The spatial correlation may be measured by the Global and Local Moran's I, with the former reflecting the spatial correlation as a whole, and the latter measuring the local correlation. The global correlation may be used to describe the overall spatial correlation status of air pollution in the BTH "2+26" cities, and determine whether air pollution is spatially aggregated in nature. The Moran's I values for $SO_2$, $PM_{2.5}$, $PM_{10}$, $NO_x$, and VOCs were all positive and were significant at the 5% significance level (Table 7). This shows that the overall pollution emission from industrial enterprises was significantly positively correlated with the spatial distribution of the BTH "2+26" cities; air pollution demonstrated a spatial agglomeration effect in the BTH "2+26" cities. This meant that the pollutant emission sources among BTH APTCC cities affect each other, as does the pollution concentration of $SO_2$, $NO_x$,

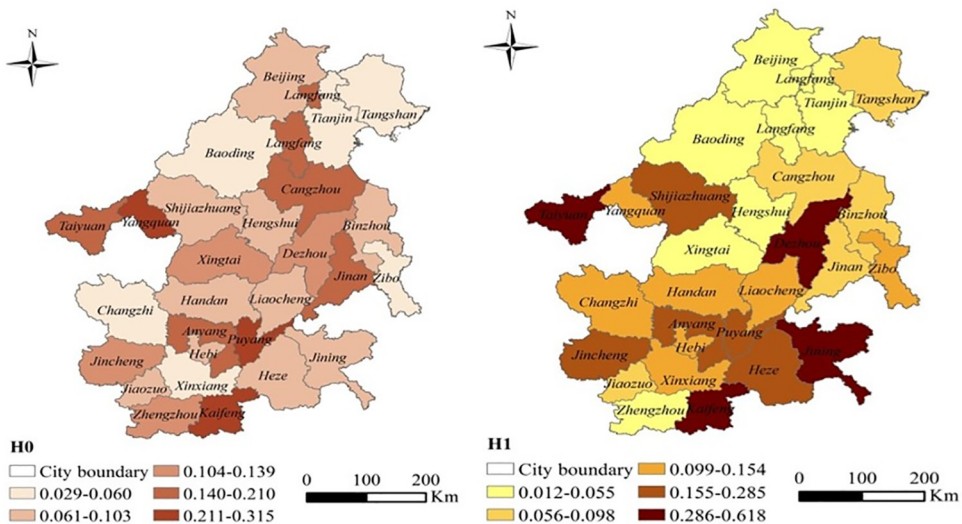

**Fig 3. Pollutant emission agglomeration in the BTH "2+26" cities.** The map was prepared in ARCGIS using political boundaries from the Global Database of Administrative Areas (https://resources.arcgis.com/en/help/).

$PM_{2.5}$, and $PM_{10}$ in adjacent cities; this confirmed the conclusions of previous studies [67–69].

**3.4.2 Local spatial autocorrelation test.** Moran scatter plots were able to describe local spatial correlations in the BTH "2+26" cities. The spatial autocorrelation analysis of $SO_2$ emissions demonstrated that 74% of the county-level pollutant emissions was positively spatially correlated, and $NO_x$, $PM_{2.5}$, and VOC emissions had positive spatial correlation ratios of 74%, 74%, and 68%, respectively. More than 50% of the polluting district and county agglomerations were located in the first and third quadrants; these are characterized by the "High-High" or "Low-Low" types. This indicates that there was a significant spatial autocorrelation in the pollutant emissions of industrial enterprises in the BTH "2+26" cities in 2016. Pollution agglomeration in most districts and counties was similar to that in neighboring counties. The districts and counties with high levels of pollution discharge were geographically adjacent, and the districts and counties with low pollution discharge agglomeration were also largely in close proximity to each other.

Anselin and Florax [70] proposed the local indicators of spatial association (LISA); analysis of the LISA cluster map may further reflect the spatial clustering significance of pollutant emissions in the BTH "2+26" cities. We established the LISA agglomeration map of different pollutant emissions of the BTH "2+26" cities (Fig 4). The Moran scatter plot was divided into four quadrants according to the four types of pollution control performance occurring in each region: high-high (H-H), low-high (L-H), low-low (L-L), and high-low (H-L), which correspond to the first, second, third, and fourth quadrants, respectively. The first quadrant represents the significant concentration of high pollutant emissions, and the third quadrant

**Table 7. Global Moran's I statistic for pollutant emissions from industrial enterprises.**

|  | SO₂ | NOₓ | PM₂.₅ | VOC |
|---|---|---|---|---|
| Moran's I | 0.320 | 0.316 | 0.296 | 0.285 |
| Z value | 8.905 | 8.614 | 8.054 | 7.994 |
| P value | 0.000 | 0.000 | 0.000 | 0.000 |

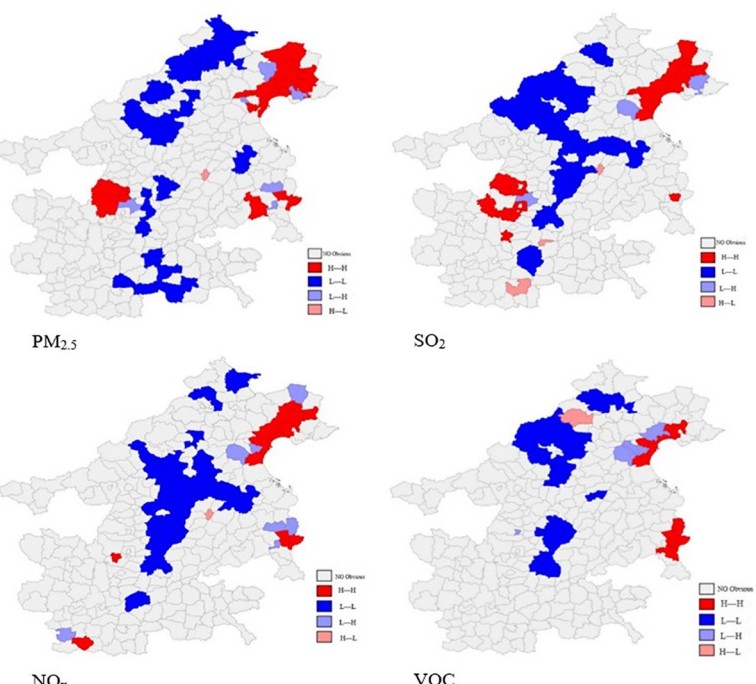

**Fig 4. Local Indicators of Spatial Association (LISA) analysis of local spatial autocorrelation of different pollutants in the BTH "2+26" cities.**

represents the significant concentration of low pollutant emissions. The results show that the main high-value agglomeration areas occurred in Tianjin, Tangshan, and the eastern coastal areas of Hebei. The low-value agglomeration areas mainly occurred in the central region, including Beijing, Baoding, Cangzhou, Hengshui, Xingtai, and Handan.

## 3.5 The impact of corporate agglomeration on air pollution

**3.5.1 Correlation analysis of enterprise agglomeration and pollution agglomeration.** We calculated the Pearson correlation coefficient between enterprise agglomeration and pollutant emission agglomeration. The results showed that the correlation coefficient between enterprise agglomeration and pollutant emission agglomeration was 0.631; this is significantly positive at the 1% level, indicating that there is a strong positive correlation between enterprise agglomeration and pollutant emission agglomeration in the "2+26" cities. The high concentration of enterprises in the spatial distribution means that the pollution discharge in this area is also highly concentrated. The Pearson correlation between enterprise agglomeration and pollution agglomeration is shown in Table 8.

**3.5.2 Analysis of the impact of enterprise agglomeration on air quality.** A spatial measurement model was established to determine the relationship between air quality, enterprise

**Table 8. Pearson correlation of enterprise agglomeration and pollution agglomeration.**

| | Enterprise agglomeration | Pollutant emission agglomeration |
|---|---|---|
| Enterprise agglomeration | 1.000 | 0.631*** |
| Pollutant emission agglomeration | 0.631*** | 1.000 |

Note: *** denotes the significant correlation at 0.01 (bilateral).

**Table 9. The descriptive statistics of variables.**

| Variable name | Unit | Mean | Standard Deviation | Minimum | Maximum |
|---|---|---|---|---|---|
| AQI | / | 111.7 | 10.17 | 92.4 | 132.4 |
| EGI | / | 0.027 | 0.055 | 0.0002 | 0.284 |
| perGDP | Yuan | 55,113 | 25,957 | 19,521 | 118,189 |
| POP | person/km$^2$ | 699.87 | 279.71 | 246.25 | 1324.05 |
| IND | / | 0.474 | 0.082 | 0.193 | 0.652 |

Note: The air quality index (AQI) and industrial structure (IND) are dimensionless units, the unit of per capita GDP (per GDP) is RMB yuan, and the unit of population density (POP) is (person/km$^2$).

agglomeration, and economic development. In this spatial econometric model, the daily average of the AQI in 2016 was the explanatory variable, and the EGI index was used as the relevant indicator for agglomeration. Other control variables include per capita GDP (perGDP), population density (POP), and industrial structure (IND). The descriptive statistics of the variables is provided in Table 9.

To determine whether the SEM or the spatial lag (SAR) model was more suitable for the sample of this study, we initially conducted the ordinary least squares (OLS) estimation to observe the Lagrangian multiplier lag (LM), error, and robustness tests. The test contains two statistics: LM-Error and LM-Lag. If these two statistics are not significant, the OLS model is selected as the final model; if the robust LM-Error statistic is significant, it points to the SEM; and if the robust LM-Lag statistic is significant, it points to the SAR. The results of the OLS estimation and robustness tests are shown in Table 10.

The table shows that LM (lag) and LM (error) were not significant, indicating that it is not applicable to the SAR or SEM models. Additionally, the coefficient of the EG index was negative and insignificant. This result is inconsistent with those of previous spatial autocorrelation tests. Therefore, considering a possible non-linear relationship, we used the square of EG as the control variable for least squares regression.

The results of the OLS regression show that the LM (error) result is statistically more significant than the LM (lag); both RLM (lag) and RLM (error) were not significant. Therefore, based on the model judgment criteria, an SEM was selected for further testing. We used the structural equation modeling estimation, where the results are shown in Table 11.

**Table 10. Estimation of the impact of enterprise agglomeration on the air pollution of OLS1 and OLS2.**

| Variable | OLS1 | OLS2 |
|---|---|---|
| CONSTANT | 5.7663*** | 11.2441*** |
| EG | -0.4204 | -4.1516*** |
| EG_2 | / | 12.7351*** |
| lnperGDP | -0.1118** | -0.0972* |
| POP | 0.0002** | 0.0002** |
| IND | 0.0946 | -0.1281 |
| LM (lag) | 1.8766 | 1.8824 |
| RLM (lag) | 0.0357 | 0.2033 |
| LM (error) | 2.385 | 4.3403** |
| RLM (error) | 0.544 | 2.6612 |

Note: "*", "**", and "***" are represented at 10%, 5%, and 1%, respectively.

**Table 11. SEM estimation of the impact of enterprise agglomeration on air pollution.**

| Variable | OLS |
|---|---|
| ρ | 0.4887*** |
| CONSTANT | 5.3653*** |
| EG | -2.1623*** |
| EG_2 | 6.3568** |
| lnperGDP | -0.0737* |
| POP | 0.0002*** |
| IND | 0.0883 |

Note: "*", "**", and "***" are represented at 10%, 5%, and 1%, respectively.

In industrial agglomeration areas, there is often a large scale of production that attracts labor and increases population density, resulting in more severe air pollution. However, in the post-industrial period, a service-oriented city attracts an increasing number of people and causes air pollution. The analysis results of the SEM found that the spatial error regression coefficient was highly statistically significant. This confirms the spatial correlation of air pollution in the BTH "2+26" cities and the reasonability of developing a spatial measurement model. The value of the spatial error regression coefficient was positive, indicating that air pollution has strong spatial dependence and there are spatial spillover effects. For the BTH APTCC, air pollution in one area induced air pollution in adjacent areas. As such, this spatial spillover effect indicates that the level of air pollution in a region is affected by the economic development within a region, demographic changes, and industrial enterprise agglomeration. It is also impacted by the economic development, demographic changes, and industrial enterprise agglomeration level of its neighboring regions.

The primary and secondary terms of the EG index passed the 1% significance level test, and the primary coefficient $\beta_1$ was negative, while the secondary coefficient $\beta_2$ was positive, indicative of an inverted, U-shaped relationship between enterprise agglomeration and air quality in the BTH "2+26" cities. First, as enterprise agglomeration grows, the scale of production increases, usually attracting more labor. With economic development, the industrial structure is continuously adjusted, and air quality is improved. However, as agglomeration increases excessively and the scale of production blindly expands, this generates an explosion in population growth. As such, it reaches a threshold where any further growth in agglomeration leads to deteriorating air quality. Alternatively, if the industrial structure is not well adjusted as the enterprises agglomerate blindly, air quality will deteriorate once it passes an inflection point [71–73].

This type of industrial agglomeration has an inverted, U-shaped relationship with environmental quality, which conforms to the characteristics of the Environmental Kuznets Curve [37], which describes the relationship between economic development and pollution. It postulates that as the economy develops, pollutant emissions increase until a certain threshold is reached; then, the high economic level contributes to technological innovations that aim to relieve pollution emissions, causing emissions to decrease. Therefore, there is an inverted, U-shaped curve between economic development and environmental pollution. This indicates the importance of promoting the rationalization of the industrial structure and establishing a long-term mechanism for environmental governance to effectively balance economic growth and environmental protection [74].

## 4. Conclusions

The relationship between industrial agglomeration and environmental pollution is complex. Previous studies on industrial agglomeration largely used city-level panel data. This study used

the EGI, HHI, and spatial correlation analysis to analyze the spatial emission and agglomeration characteristics of pollution sources. This was carried out using data on 73,353 enterprises in the BTH "2+26" cities of China; we were able to provide new evidence of the impact of industrial agglomeration on environmental performance.

Through the calculation of the adjusted EGI, we found that there were mainly two types of cities with relatively high concentrations of enterprises. The first category includes the areas surrounding Beijing and Tianjin, excluding the two cities themselves. Due to changes in government policies on urban and industrial development, enterprises have relocated from the core areas of Beijing to its peripheries; the transferred area surrounds the urban regions of Beijing and Tianjin. The other category includes cities such as Taiyuan, Zhengzhou, and Jinan, which are provincial capitals with a relatively dense distribution of industries; a large number of these industries are resource-intensive, forming scaled economies and a certain level of agglomeration. As these cities have not begun to evacuate or transfer industries, they are still considered economic centers. These two types of agglomeration characteristics represent differences in the development models and stages in different regions of China.

The results of Global and Local Moran's I showed that the overall pollution discharge of industrial enterprises was significantly positively correlated with the spatial distribution of the BTH "2+26" cities. Air pollution presented a spatial agglomeration effect, meaning that pollutant emissions in any of the BTH "2+26" APTCC will impact other surrounding areas. Additionally, the pollution concentrations of $SO_2$, $NO_x$, $PM_{2.5}$, and $PM_{10}$ in adjacent cities will also affect nearby cities, which verifies the conclusions of previous studies. We should be alert to the problem of industrial cliffs caused by excessive industrial gradients among the BTH "2 +26" APTCC. Gradually, a cross-complementary and distinctive regional division of labor pattern should be formed, ultimately promoting the coordinated development of the BTH region [75].

There is a strong positive correlation between enterprise agglomeration and pollutant emission agglomeration in the BTH "2+26" cities. Through spatial measurement model analysis, we found that there was an inverted U-shaped relationship between the industrial enterprise agglomeration and air quality in the BTH. This is likely due to the increase in the agglomeration of enterprises; the scale effect or an industrial chain gradually forms between enterprises, introducing improvements in the technological level and energy saving. However, when the agglomeration reaches a certain point, the diffusion effect of pollution causes further serious air pollution to the surrounding areas as agglomeration proceeds.

Industrial agglomeration is inseparable from different forms of pollution. The spatial distribution of enterprises, such as the formation of industrial parks, may enhance air quality to a certain extent. However, urban pollution control needs to be combined with the optimization of local urban and industrial layouts. It is also necessary to consider the industrial development of nearby cities and the impact of pollutant emissions on the air quality of adjacent cities.

This study analyzed the relationship between atmospheric pollution and industrial agglomeration from the BTH APTCC and provided evidence for the diffusion effect of air pollution in urban agglomerations. These findings may be applied in air pollution management in urban agglomerations such that it is more objective, reliable, and powerful. It is recommended that future studies use more detailed data and related data mining technology to explore air pollution characteristics in different areas.

## Supporting information

**S1 Data.**
(XLSX)

## Author Contributions

**Conceptualization:** Cuicui Xiao, Jingbo Zhou.

**Data curation:** Cuicui Xiao, Shumin Zhang.

**Formal analysis:** Cuicui Xiao.

**Investigation:** Cuicui Xiao.

**Methodology:** Cuicui Xiao, Jingbo Zhou, Shumin Zhang.

**Resources:** Xin Wang.

**Supervision:** Cuicui Xiao, Jingbo Zhou.

**Visualization:** Cuicui Xiao, Shumin Zhang.

**Writing – original draft:** Cuicui Xiao.

**Writing – review & editing:** Cuicui Xiao.

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
