## [Decision Letter · Decision Letter 0]

20 Jan 2021

PONE-D-20-38132

Industrial agglomeration and air pollution: A new perspective from enterprises of Beijing-Tianjin-Hebei in China

PLOS ONE

Dear Dr. Zhou,

Thank you for submitting your manuscript to PLOS ONE. After careful consideration, we feel that it has merit but does not fully meet PLOS ONE’s publication criteria as it currently stands. Therefore, we invite you to submit a revised version of the manuscript that addresses the points raised during the review process.

We look forward to receiving your revised manuscript.

Kind regards,

Chon-Lin Lee, Ph.D.

Academic Editor

PLOS ONE

Journal Requirements:

'This study was supported by the Humanities and Social Sciences Project Youth Fund of Ministry of Education in China (18YJCZH196), the Fundamental Research Funds for the Central Universities (FRF-TP-19-069A1) “Application Research of Policy Analysis Methods and Techniques in Environmental Fields under the Background of Big Data” and Emission Characteristics and Total Amount Estimation of Non-key Industrial Enterprises (20170118)..'

'The author(s) received no specific funding for this work.'

3. We note that Figures 1-7 and 9 in your submission contain map images which may be copyrighted.

a. You may seek permission from the original copyright holder of Figures 1-7 and 9 to publish the content specifically under the CC BY 4.0 license. 

5. Please ensure that you refer to Figure 8 in your text as, if accepted, production will need this reference to link the reader to the figure.

6. We note you have included tables to which you do not refer in the text of your manuscript. Please ensure that you refer to Tables 7 and 8 in your text; if accepted, production will need this reference to link the reader to each Table.

7. Please include captions for your Supporting Information files at the end of your manuscript, and update any in-text citations to match accordingly. Please see our Supporting Information guidelines for more information: http://journals.plos.org/plosone/s/supporting-information

Reviewers' comments:

Reviewer's Responses to Questions

**Comments to the Author**

1. Is the manuscript technically sound, and do the data support the conclusions?

Reviewer #1: Yes

Reviewer #2: Partly

2. Has the statistical analysis been performed appropriately and rigorously? 

Reviewer #1: Yes

Reviewer #2: Yes

3. Have the authors made all data underlying the findings in their manuscript fully available?

Reviewer #1: Yes

Reviewer #2: Yes

4. Is the manuscript presented in an intelligible fashion and written in standard English?

Reviewer #1: Yes

Reviewer #2: Yes

5. Review Comments to the Author

Reviewer #1: The objective of this study is to investigate the impacts of industrial agglomeration on air pollution and to compare the spatial emission characteristics of industrial pollution sources in Beijing-Tianjin-Hebei (BTH) areas. By applying kernel density to analyze density distribution of industrial agglomeration, EGI, HHI, and spatial autocorrelation Global Moran’s I, this study found that the pollutant enterprises were agglomerated at the county level in the “2+26” cities of BTH. Further, the results of EGI, HHI, and spatial autocorrelation analysis has also been discussed in this study. In addition, as an improvement, some comments below need to be considered.

1. Proofread might be helpful to improve scientific writing.

2. Abstract (line 3, page 1), “with the highest PM2.5 concentration", this sentence seems to confirm that this study only focused on exposure to PM2.5 exposure. However, in further discussion this study discussed several types of pollutants (i.e. PM2.5, SO2, NOx, VOC, etc.). I suggest improving this information which supports the main topic (air pollution in general).

3. Abstract (line 6, page 1), “analyses”, I only found one method (Global Moran's I), so, it should be singular “analysis”.

Furthermore, I think Global Moran's I is better known as “Spatial Autocorrelation” not “Spatial Correlation”, please clarify.

4. Data source and description (page 7), The period of all datasets used needs to be clarified.

5. Regarding the area unit, is "county" the same as "city"? if not, related to the Ellison and Glaeser Index estimation, knowing that this analysis used county (district) level data, I suggest describing the total number of counties analyzed.

6. (line 7, page 14) where does the estimate > 85% come from?

7. (line 13, page 14) “… there are two main types of cities with a high concentration of enterprises”, In connection with this explanation, is there a threshold or standard to categorize the type of city (low-high concentration of enterprises) based on the EGI?

8. Local spatial autocorrelation test (page 24), why this analysis only showed the results of PM2.5, SO2, NOx, and VOC, while other pollutants such as NO2, PM10, AQI, O3, and CO did not present?

9. (lines 15 - 19, page 27), “The study developed a spatial econometric model………. industrial structure (IND)”, In my opinion, it would be better if this explanation describes in the materials and methods section.

10. In general, especially in the discussion of the results section, this part lacks references.

Thus, the study findings do not provide information related support or comparisons with previous studies.

11. Conclusions (pages 30 – 32), I suggest briefly explain the conclusion point-by-point

Reviewer #2: It is an interesting attempt to apply economic formulas such as industrial clustering to air pollution, and I like the short and clean style of the writing. However, both the cited literatures and the article itself are still focused on economic discussions, please give more discussion to the characteristics or effects of individual pollutants, such as PM2.5, NOx, SO2, or VOC etc. In addition, it is good to uses many economics models as a kaleidoscope, but please give more narrative to the data sources or parameter settings, and it would be better to compare with the result from different methods.

The following questions are specific to the manuscript.

1 In the literature review, please add descriptions of the individual air pollutants cited, and the models and methods used in this study as background knowledge.

2 About the clustering index, Ellison and Glaeser index (EGI) and Herfindahl Index (HHI), (a) in Section 4.2, EGI values was only applied in the city, please supplement the EGI values of the clusters in the industry categories listed in Table 4 and discuss with the cross-correlation with cities; (b) both the Herfindahl Index (HHI) and EGI can be applied as the index for clustering. Please explain the difference between the result of Herfindahl Index (HHI) in Table 6 and the EGI in Table 3 Why not use the HHI in Table 6 to further calculate the EGI?

3 About spatial measurement model, (a) in Section 3.3, please discuss the selection principles of the variables and the validation method of the model; (b) AQI is including O3, PM2.5, PM10, CO, SO2, and other sub-indicators. Also, the calculation of EGI was including HHI. Is it appropriate to use them together in the Table 7? Please check it by VIF or other collinearity diagnostics; (c) in Section 4.5.2, per “this result is inconsistent with the previous spatial autocorrelation test results. Therefore, considering the possible non-linear relationship, the square of EG is used as the control variable for least squares regression.” EGI is inconsistent with the test results, and still use the square of EGI as variable. The model becomes quadratic equation and more complicated. Is it appropriate using it? and please explain what the meaning of the square of EGI.

4 NO2 was mentioned in section 3.2 and applied in Table 2 and Table 7 (Moran), but in Table 5, Figure 4 (Kernel density analysis), Table 6 (HHI), Figure 8 (Moran scatterplots) and Figure 9 (LISA) the emission is using NOx as index. Please explain the criteria of selecting NO2 or NOx.

5 The result of each index or model was not similar at all. For example, compared with the results of EGI in Figure 2, Beijing, Yangquan, Puyang, and Kaifeng have the highest concentration of EGI index, but with Kernel density analysis, the concentration is at the county level, mainly in the three major agglomerations. Please discuss and compare in depth the differences between the result of Figure 2 (EGI), Figure 3 to Figure 6 (Kernel density analysis), Figure 7 (HHI), and Figure 9 (LISA), and Table 6 HHI of each pollutant.,

6 In section 4.4.1, “from the perspective of correlation strength, the correlation of each pollutant is greater than 0.2.” Is 0.2 a critical value in other publication?

7 In section 4.5.2, (a) please clarify the SEM abbreviation is a “spatial error model” or a “structural equation modeling”; (b) please add the descriptions of OLS, SEM, SAR, LM, and RLM in the methodology; (c) please add the description and explanation of the “threshold” (turning point) for the "U-shaped" relationship”.

Some minor suggestions:

1. In the literature review, (a) Leeuw (2001) and Frank (2001) are duplicated; (b) the format of Anselin (1995), Porter (1988), Ramón (2000), Yan (2011), Wang (2017), Zhang (2012) is not consistent with others, please check the format again.

2. In Section 3.2, (a) please add more description to “transmission channel cities” and “surrounding areas”; (b) please add the information of the air quality monitoring stations.

3. In Section 3.3, (a) please add the value of “h” and “bi” used in the kernel density estimation; (b) please separate the Herfindahl index into a separate section and discuss it separately; (c) please add the description of the Local Indicators of Spatial Association (LISA) method.

4. In Section 4.2, (a) the total number of 44 industries in Table 4 is 74017, but in Section 3.2 it is 73353. Is it correct? Please confirm with the number; (b) please add the data source of emission data in Table 5; (c) there are only five industries in Table 5, please explain the standards to choose the industries.

5. In Section 4.4.2, (a) please add the MORAN index of VOC in Table 7; (b) please add the description of Moran scatterplots (Figure 8) in method; (c) the MORAN index of SO2 in Figure 8 is 0.3093 and the MORAN index of SO2 in Table 7 is 0.565. Is it correct? Please confirm with it.

6 In Section 4.5.1, please specify the pollution emission in Table 8 and the units in Table 9. In addition, please add the data source of these two tables.

6. PLOS authors have the option to publish the peer review history of their article (what does this mean?). If published, this will include your full peer review and any attached files.

Reviewer #1: No

Reviewer #2: No

---

## [Author Response · Author response to Decision Letter 0]

1 Apr 2021

Response to the Editors and reviewers 

1.Responses to the Editors:

Comment (1): Please ensure that your manuscript meets PLOS ONE's style requirements, including those for file naming.

Response: Thank you for your kind advice, and details of this article have been improved. I would be grateful if the revised manuscript was accepted by PLOS ONE.

Comment (2): The funding information should not appear in the Acknowledgments section or other areas of your manuscript.

Response: Thanks for your comments. We have removed the funding information in the Acknowledgments section.

Comment (3): We note that Figures 1-7 and 9 in your submission contain map images which may be copyrighted. All PLOS content is published under the Creative Commons Attribution License (CC BY 4.0), we cannot publish previously copyrighted maps or satellite images created using proprietary data, such as Google software (Google Maps, Street View, and Earth).

Response: Thanks for your kind advice and comments. We have revised and increased the source information of maps, the maps were prepared in ARCGIS using political boundaries from the Global Database of Administrative Areas (https://resources.arcgis.com/en/help/).

Comment (4): Please ensure that you have an ORCID iD and that it is validated in Editorial Manager. 

Response: Thank you for your kind advice. We have updated our information in the ORCID, and the ORCID account name is zhoujb@ruc.edu.cn. 

Comment (5): Please ensure that you refer to Figure 8 in your text as, if accepted, production will need this reference to link the reader to the figure. 

Response: Thanks for your comments. We have revised this part and removed Figure 8 in the revised version.

2.Responses to the Reviewer #1 

Comment (1): The objective of this study is to investigate the impacts of industrial agglomeration on air pollution and to compare the spatial emission characteristics of industrial pollution sources in Beijing-Tianjin-Hebei (BTH) areas. By applying kernel density to analyze density distribution of industrial agglomeration, EGI, HHI, and spatial autocorrelation Global Moran’s I, this study found that the pollutant enterprises were agglomerated at the county level in the “2+26” cities of BTH. Further, the results of EGI, HHI, and spatial autocorrelation analysis has also been discussed in this study. In addition, as an improvement, some comments below need to be considered.

Response: Thank you for your compliment, and details of this article have been improved. We will continue to modify and improve the analysis process and conclusions. I would be grateful if the revised manuscript was accepted by PLOS ONE. 

Comment (2): Proofread might be helpful to improve scientific writing.

Response: Thank you for your kind advice. We have revised the writing and details of this article have been improved.

Comment (3): Abstract (line 3, page 1), “with the highest PM2.5 concentration", this sentence seems to confirm that this study only focused on exposure to PM2.5 exposure. However, in further discussion this study discussed several types of pollutants (i.e. PM2.5, SO2, NOx, VOC, etc.). I suggest improving this information which supports the main topic (air pollution in general). 

Response: Thank you for your comments. We have rewritten the abstract, “air quality has gradually improved in China in recent years, but the Beijing–Tianjin–Hebei (BTH) region is still the most polluted area in China with the worst Air Quality Index (AQI)”. The “Technical Regulations on Ambient Air Quality Index” issued by the Ministry of Ecology and Environment stipulates that the Air Quality Index (AQI) is a dimensionless index that quantitatively describes air quality, integrating PM2.5, PM10, SO2, NOx, O3 and other indicators.

Comment (4): Abstract (line 6, page 1), “analyses”, I only found one method (Global Moran's I), so, it should be singular “analysis”.Furthermore, I think Global Moran's I is better known as “Spatial Autocorrelation” not “Spatial Correlation”, please clarify. 

Response: Thank you for your kind advice, we apologize for our spelling mistakes, we have corrected the problems ,the“analyses” should be singular “analysis”， “Spatial Correlation” is also corrected as “Spatial Autocorrelation” , and details of this article have been improved.

Comment (5): Data source and description (page 7), The period of all datasets used needs to be clarified. 

Response: Thanks for your question and suggestion. We have rewritten this part. The research data includes the basic information (such as company name, geographic location, industry type) and pollution emission of 73,353 industry enterprises, we calculated the percentages from it. The data used in this study includes three categories: pollution emissions data of enterprises, the urban economic and social development data, and air quality data. The statistical scope of all data is 2016. The micro data used comes from the “Beijing-Tianjin-Hebei Air Pollution Transmission Channel Pollution Source Emission List” issued by the Ministry of Ecology and Environment in 2017. The urban economic and social development data (including total industrial output value, industrial structure, population density, and per capita GDP) are derived from the “China City Statistical Yearbook” issued by the National Bureau of Statistics in 2017, and urban air quality data in the Beijing-Tianjin-Hebei region (including PM2.5, PM10, SO2, NO2) are collected from the National Urban Air Quality Real-time Release Platform of the Ministry of Ecology and Environment (http://106.37.208.233:20035/), and the data period is 2016.

Comment (6): Regarding the area unit, is "county" the same as "city"? if not, related to the Ellison and Glaeser Index estimation, knowing that this analysis used county (district) level data, I suggest describing the total number of counties analyzed. 

Response: Thanks for your question and suggestion. The minimum unit of calculation here is the county (district) level. The county or district is not same as city, there are 303 districts and counties in the BTH “2+26” cities. We have revised this part, and details of this part have been improved.

Comment (7): (line 7, page 14) where does the estimate > 85% come from?

Response: Thank you for your comments. We classified and make the industry categories of enterprises statistics according to “the National Economic Industry Classification GBT 4754-2017”. It was found that the surveyed enterprises were distributed in 44 industries (Table 4). Among them, the manufacturing enterprises accounting for more than 85%, including the nonmetallic mineral products industry, metal products, general equipment manufacturing, rubber and plastic products, etc. The data comes from the environmental statistics of the China National Environmental Monitoring Centre (Due to data confidentiality, the research data has not been disclosed in this study). The research data includes the basic information of pollution source industry enterprises, pollution source emission inventory, number of enterprises, enterprise industry category, spatial geographic location, pollution emissions, etc. If they want to obtain the research data, they need to seek the consent of China National Environmental Monitoring Centre.

Comment (8): (line 13, page 14) “… there are two main types of cities with a high concentration of enterprises”, In connection with this explanation, is there a threshold or standard to categorize the type of city (low-high concentration of enterprises) based on the EGI? 

Response: Thank you for your comments. We have revised this part, added the standard to categorize the EGI. According to Ellison and Glaeser, When the EGI value is less than 0.02, it is considered as low agglomeration; when the EGI is greater than 0.02 and less than 0.05, it is considered as moderate agglomeration; when the EGI is greater than 0.05, which indicates that the industrial agglomeration in this region is relatively high.

Comment (9): Local spatial autocorrelation test (page 24), why this analysis only showed the results of PM2.5, SO2, NOx, and VOC, while other pollutants such as NO2, PM10, AQI, O3, and CO did not present? 

Response: Thank you for your question. We apologize for our mistakes on the spatial correlation. We have rewritten this part. Table 7 gives the Global Moran’s I statistic for SO2, NOx, PM2.5, VOC pollution emissions from industrial enterprises and the LISA analysis also showed the results of PM2.5, SO2, NOx and VOC in the revised version.

Comment (10): (lines 15 - 19, page 27), “The study developed a spatial econometric model………. industrial structure (IND)”, In my opinion, it would be better if this explanation describes in the materials and methods section. 

Response: Thank you for your kind advice. We have added the spatial econometric model in the materials and methods section, details of this article have been improved.

Comment (11): In general, especially in the discussion of the results section, this part lacks references. Thus, the study findings do not provide information related support or comparisons with previous studies. Conclusions (pages 30 – 32), I suggest briefly explain the conclusion point-by-point.

Response: Thank you for your kind advice. We rewrite the results and discussion section, explained and compared the conclusions point-by-point, verified our conclusions, and supplemented a lot of literatures and references in this part of results and discussion, and give some comparisons with previous studies.

3.Responses to the Reviewer #2 

Comment (1): It is an interesting attempt to apply economic formulas such as industrial clustering to air pollution, and I like the short and clean style of the writing. However, both the cited literatures and the article itself are still focused on economic discussions, please give more discussion to the characteristics or effects of individual pollutants, such as PM2.5, NOx, SO2, or VOC etc. In addition, it is good to uses many economics models as a kaleidoscope, but please give more narrative to the data sources or parameter settings, and it would be better to compare with the result from different methods.

Response: Thank you for your compliment, and details of this article have been improved. I would be grateful if the revised manuscript was accepted by PLOS ONE. The original literature review mainly focused on the relationship between industrial agglomeration and environmental pollution, and the environmental pollution situation in the Beijing-Tianjin-Hebei (BTH) region. The revised version added and discussed the pollution characteristics of different pollutants in BTH region. We have also added the data source and parameter settings, we have also compared and analyzed the applicable scope of EGI and HHI. 

Comment (2): In the literature review, please add descriptions of the individual air pollutants cited, and the models and methods used in this study as background knowledge. 

Response: Thank you for your question and suggestion. We have added descriptions of the individual air pollutants in the literature review. The revised version added and discussed the models and methods used in this study. Details have been improved in this article.

Comment (3): About the clustering index, Ellison and Glaeser index (EGI) and Herfindahl Index (HHI), (a) in Section 4.2, EGI values was only applied in the city, please supplement the EGI values of the clusters in the industry categories listed in Table 4 and discuss with the cross-correlation with cities; (b) both the Herfindahl Index (HHI) and EGI can be applied as the index for clustering. Please explain the difference between the result of Herfindahl Index (HHI) in Table 6 and the EGI in Table 3 Why not use the HHI in Table 6 to further calculate the EGI? 

Response: Thank you for your comment and suggestion. We revised the conclusion part to a certain extent. We used Ellison–Glaeser index (EGI), Herfindahl-Hirschman Index (HHI), and Spatial correlation analysis to analyze the spatial emission and agglomeration characteristics of the industrial pollution sources in different cities. The EGI was used to analyze the degree of industrial enterprises agglomeration. EG index is an agglomeration index proposed by Elision & Glaeser (1997) to measure the degree of industrial agglomeration. In order to quantitatively analyze the degree of agglomeration of enterprises in different industries, and explore the relationship between enterprise agglomeration and air pollution, it is necessary to select appropriate measurement indicators. Taking into account that large number of practices of the EGI in measuring industrial agglomeration, and the regional economic indicators are included in the calculation, we thought EGI is suitable for the comparison of agglomeration among different regions, therefore, the EGI is selected to measure the agglomeration of enterprises.

Compared with the Ellison-Glaeser index, the Herfindahl index is simple to calculate and can reflect changes in industry or corporate market concentration market monopoly more accurately. Since we obtained the pollution emission data of 73, 353 micro-enterprises, and the Herfindahl index can better measure the pollution agglomeration of micro-enterprises in the region, the Herfindahl index is selected to measure the agglomeration of pollution emissions.

Comment (4): About spatial measurement model, (a) in Section 3.3, please discuss the selection principles of the variables and the validation method of the model; (b) AQI is including O3, PM2.5, PM10, CO, SO2, and other sub-indicators. Also, the calculation of EGI was including HHI. Is it appropriate to use them together in the Table 7? Please check it by VIF or other collinearity diagnostics. 

Response: Thanks for your question. we apologize for our mistakes in Table 7, we rewrote this part and details of this article have been improved. In order to find the relationship between air quality, enterprise agglomeration and economic development level, we established a spatial measurement model. In this spatial econometric model, the daily average of the Air Quality Index (AQI) in 2016 is explanatory variable, and the EGI index is used as the relevant indicator EG for agglomeration. Other control variables include per capita GDP (per GDP), population density (POP) and industrial structure (IND). In the spatial measurement model, we did not use O3, PM2.5, PM10, SO2 , CO index and HHI.

Before establishing the spatial measurement model, in order to determine whether the spatial error (SEM) or the spatial lag (SAR) model is more suitable for the sample of this study, we first performed the traditional least square method (OLS) estimation to observe the Lagrangian multiplier lag (LM), Error and Robustness test. The test contains two statistics: LM-Error and LM-Lag. If these two statistics are not significant, the OLS model is selected as the final model, if the Robust LM-Error statistic is significant, it points to the spatial error model, and if the Robust LM-Lag statistic is significant, then it points to the spatial lag model. 

We have checked the independent variables in the model by VIF, and the results showed that the variables are independent.

Comment (5): (c) in Section 4.5.2, per “this result is inconsistent with the previous spatial autocorrelation test results. Therefore, considering the possible non-linear relationship, the square of EG is used as the control variable for least squares regression.” EGI is inconsistent with the test results, and still use the square of EGI as variable. The model becomes quadratic equation and more complicated. Is it appropriate using it? and please explain what the meaning of the square of EGI.

Response: Thanks for your question. we rewrote this part and explained the Spatial measurement model analysis.Consistent with the comments (5), in order to find the relationship between air quality, enterprise agglomeration and economic development level, we established a spatial measurement model. In this spatial econometric model, the daily average of the air quality index (AQI) in 2016 is explanatory variable, the EGI index is used as the relevant indicator EG for agglomeration. Other control variables include per capita GDP (perGDP), population density (POP), and industrial structure (IND). In the spatial measurement model,we did not use O3,PM2.5,PM10,SO2 and CO these index. In the model, we found that LM (lag) and LM (error) are not significant, indicating that it is not suitable for spatial lag model or spatial error model, and the coefficient of EG index is negative and not significant. This is inconsistent with the theoretical research and the previous spatial autocorrelation test. Therefore, considering the possible non-linear relationship, the variable EG^2 is added to obtain the results of the study, but we did not try a cubic specification. Through variable EG^2 in the OLS2, both the primary and secondary terms of the EG index pass the 1% significance level test, and the primary coefficient β1 is negative while the secondary coefficient β2 is positive, which indicates a “inverted U-shaped” relationship between the enterprise agglomeration and air quality in BTH “2+26” cities. This kind of industrial agglomeration has an inverted U-shaped relationship with environmental quality, which conforms to the characteristics of the environmental Kuznets curve (EKC).

Comment (6): NO2 was mentioned in section 3.2 and applied in Table 2 and Table 7 (Moran), but in Table 5, Figure 4 (Kernel density analysis), Table 6 (HHI), Figure 8 (Moran scatterplots) and Figure 9 (LISA) the emission is using NOx as index. Please explain the criteria of selecting NO2 or NOx. 

Response: Thank you for your kind advice, we apologize for our misunderstanding and grammar mistakes on the Fig.9 and Fig.8. We have rewritten this part, NO2 is a measurement indicator in the air quality standards issued by the Ministry of Ecology and Environment, and NOx is a measurement indicator of pollutant emissions, in the Global Moran’s I and Local Moran’s I model, we use NOx as one index to analyze the spatial correlation status of air pollution of air pollution of BTH “2+26” cities.

Comment (7): The result of each index or model was not similar at all. For example, compared with the results of EGI in Figure 2, Beijing, Yangquan, Puyang, and Kaifeng have the highest concentration of EGI index, but with Kernel density analysis, the concentration is at the county level, mainly in the three major agglomerations. Please discuss and compare in depth the differences between the result of Figure 2 (EGI), Figure 3 to Figure 6 (Kernel density analysis), Figure 7 (HHI), and Figure 9 (LISA), and Table 6 HHI of each pollutant. 

Response: Thanks for your comments. In the revised manuscript, we removed the Kernel density analysis. We revised this section, explained and compared EGI and HHI results point-by-point. We used Ellison–Glaeser index (EGI), Herfindahl-Hirschman Index (HHI), and Spatial correlation analysis to analyze the spatial emission and agglomeration characteristics of the industrial pollution sources in different cities. The EG index was used to analyze the degree of industrial enterprises agglomeration, the Herfindahl index is selected to measure the agglomeration of pollution emissions. We have calculated the Pearson correlation coefficient between enterprise agglomeration and pollution emission agglomeration. The results show that the correlation coefficient between enterprise agglomeration and pollution emission agglomeration is 0.631, which is significantly positive at the 1% level, indicating that there is a strong positive correlation between enterprise agglomeration and pollution emission agglomeration in “2+26” cities. That is, the high concentration of enterprises in the spatial distribution means that the pollution discharge in this area is also highly concentrated.

Comment (8): In section 4.4.1, “from the perspective of correlation strength, the correlation of each pollutant is greater than 0.2.” Is 0.2 a critical value in other publication? 

Response: Thanks for your comments. In the revised manuscript, we removed “from the perspective of correlation strength, the correlation of each pollutant is greater than 0.2”, 0.2 a not critical value, is an expression problem. We rewrite this part, and details of this part have been improved.

Comment (9): In section 4.5.2, (a) please clarify the SEM abbreviation is a “spatial error model” or a “structural equation modeling” (b) please add the descriptions of OLS, SEM, SAR, LM, and RLM in the methodology; (c) please add the description and explanation of the “threshold” (turning point) for the "U-shaped" relationship”.

Response: Thanks for your comments and suggestion. We clarify the SEM abbreviation is a “spatial error model”, and we add the descriptions of OLS, SEM, SAR, LM, and RLM in the methodology part, consulted and read many literatures, and explanation of the “threshold” (turning point) for the "U-shaped" relationship”. A “inverted U-shaped” relationship between the enterprise agglomeration and air quality in BTH and surrounding “2+26” cities. At first, as the enterprise agglomeration increases, the scale of production is very large, which usually attracts more labor. With the development of the economy, the industrial structure is continuously adjusted and air quality is improved. However, if enterprise agglomeration increases excessively and the scale of production blindly expands when the enterprise agglomeration reaches the threshold, resulting in a blind increase in labor and population, any further increase in the agglomeration will lead to a deterioration in air quality. That means when the agglomeration of enterprises reaches the threshold, if the industrial structure is ignored and the enterprises agglomerate blindly, the air quality will deteriorate.

Comment (10): In the literature review, (a) Leeuw (2001) and Frank (2001) are duplicated; (b) the format of Anselin (1995), Porter (1988), Ramón (2000), Yan (2011), Wang (2017), Zhang (2012) is not consistent with others, please check the format again. 

Response: Thank you for your kind advice, we apologize for our mistakes, we have corrected these problems in the literature review and details of this article have been improved.

Comment (11): In Section 3.2, (a) please add more description to “transmission channel cities” and “surrounding areas”; (b) please add the information of the air quality monitoring stations. 

Response: Thank you for your kind advice. The “Air Pollution Prevention and Control Work Plan of Beijing–Tianjin–Hebei and Surrounding Areas in 2017” defines the scope of remediation in the BTH atmospheric pollution transmission channel cities (APTCC). These include Beijing, Tianjin, Shijiazhuang, Tangshan, Langfang, Baoding, Cangzhou, Hengshui, Xingtai, Handan, Taiyuan, Yangquan, Changzhi, Jincheng, Jinan, Zibo, Jining, Dezhou, Liaocheng, Binzhou, Heze, Zhengzhou, Kaifeng, Anyang, Hebi, Xinxiang, Jiaozuo, and Puyang (i.e., the “2+26” APTCC cities) (Fig. 1). There are 160 urban air quality stations in the “2+26” APTCC cities which showed in the Fig.1. we have explained the information in the revised version and details of this article have been improved.

Comment (12): In Section 3.3, (a) please add the value of “h” and “bi” used in the kernel density estimation; (b) please separate the Herfindahl index into a separate section and discuss it separately; (c) please add the description of the Local Indicators of Spatial Association (LISA) method. 

Response: Thank you for your kind advice. In the revised manuscript, we removed the Kernel density analysis. We separated the Herfindahl index into a separate section and discuss it separately, added the description of the Local Indicators of Spatial Association (LISA) method in the method section.

Comment (13): In Section 4.2, (a) the total number of 44 industries in Table 4 is 74017, but in Section 3.2 it is 73353. Is it correct? Please confirm with the number; (b) please add the data source of emission data in Table 5; (c) there are only five industries in Table 5, please explain the standards to choose the industries. 

Response: Thanks for your comments and suggestion. We confirm that the number of enterprises in our study is 73,353, and the total number of industry enterprises in Table 4 is 74,017, including 664 other enterprises, those 664 enterprises are not included in our study. The total 73,353 industrial enterprises in BTH were divided into 11,168 key survey enterprises and 62,185 non-key survey enterprises. The micro data used comes from the “Beijing-Tianjin-Hebei Air Pollution Transmission Channel Pollution Source Emission List” issued by the Ministry of Ecology and Environment in 2017.We marked the data sources in the revised version. We obtained the pollution discharge list of the 73,353 industrial enterprises in BTH and the surrounding “2+26” cities in 2016 and statistics on the pollution emissions from different industries. In Table 5, these industries have the largest emissions, so these industries are listed in the Table.

Comment (14): In Section 4.4.2, (a) please add the MORAN index of VOC in Table 7; (b) please add the description of Moran scatterplots (Figure 8) in method; (c) the MORAN index of SO2 in Figure 8 is 0.3093 and the MORAN index of SO2 in Table 7 is 0.565. Is it correct? Please confirm with it. 

Response: Thank you for your question. We apologize for our mistakes on the spatial correlation. We have rewritten this part, added the description of Moran in method. We added the MORAN index of VOC in Table 7, Table 7 gives the Global Moran’s I statistic for SO2, NOx, PM2.5 and VOC pollution emissions from industrial enterprises and the LISA analysis also showed the results of PM2.5, SO2, NOx and VOC in the revised version. The MORAN index of SO2 is 0.3093.

Comment (15): In Section 4.5.1, please specify the pollution emission in Table 8 and the units in Table 9. In addition, please add the data source of these two tables. 

Response: Thank you for your kind advice, we add the units in Table 9, and the data source are given in the revised version. The air quality index (AQI) and industrial structure (IND) are dimensionless units, the unit of per capita GDP (perGDP) is RMB yuan, the unit of population density (POP) is (person/km2).

---

## [Decision Letter · Decision Letter 1]

4 May 2021

PONE-D-20-38132R1

Industrial Agglomeration and Air Pollution: A New Perspective from Enterprises in Atmospheric Pollution Transmission Channel Cities (APTCC) of Beijing-Tianjin-Hebei and Surrounding Areas, China

PLOS ONE

Dear Dr. Zhou,

Thank you for submitting your manuscript to PLOS ONE. After careful consideration, we feel that it has merit but does not fully meet PLOS ONE’s publication criteria as it currently stands. Therefore, we invite you to submit a revised version of the manuscript that addresses the points raised during the review process.

We look forward to receiving your revised manuscript.

Kind regards,

Chon-Lin Lee, Ph.D.

Academic Editor

PLOS ONE

Journal Requirements:

Reviewers' comments:

Reviewer's Responses to Questions

**Comments to the Author**

1. If the authors have adequately addressed your comments raised in a previous round of review and you feel that this manuscript is now acceptable for publication, you may indicate that here to bypass the “Comments to the Author” section, enter your conflict of interest statement in the “Confidential to Editor” section, and submit your "Accept" recommendation.

Reviewer #1: All comments have been addressed

Reviewer #2: All comments have been addressed

2. Is the manuscript technically sound, and do the data support the conclusions?

Reviewer #1: Yes

Reviewer #2: Partly

3. Has the statistical analysis been performed appropriately and rigorously? 

Reviewer #1: Yes

Reviewer #2: Yes

4. Have the authors made all data underlying the findings in their manuscript fully available?

Reviewer #1: Yes

Reviewer #2: Yes

5. Is the manuscript presented in an intelligible fashion and written in standard English?

Reviewer #1: Yes

Reviewer #2: No

6. Review Comments to the Author

Reviewer #1: the current version looks good to me and all of the comments were addressed. I dont have further comments.

Reviewer #2: The modified version is clearer than the original, but proofread might be helpful to improve scientific writing. Please check all the article again.

For example,

Line1-13, Font inconsistency with other parts of article.

Line 42, Line 88, Word choice of "externalities", "impact" would be better.

Line 45, generates more "pollutants", "pollution" would be better.

Line 47, "production concentration", The usage seems weird, and would be better to change it.

Line 48, 545, word choice of "spillover effect", Would it be better to replace a word or not used so many times.

Line 51-54, 399, “Beijing-Tianjin- Hebei (BTH) Collaborative Development Plan", “Opinions on Strengthening the Construction of the Key Platform for the Transfer of the Beijing-Tianjin-Hebei (BTH) Industry”, "Blue-Sky Protection Campaign" would be better written in italics.

Line 63, emission intensity per unit of land area, "per square kilometers" would be better.

Line 82-83, word choice of "agglomeration", "pollution aggregation" would be better.

Line 103, word choice of "pollutant emission", "pollution" would be better.

Line 114, Some Literature, "L" should not use capitalized.

Line 119, Some literature shows, It should be plural and no "s".

Line 239, Spatial correlation analysis versus Line 438, Spatial autocorrelation test, Uniform usage is better.

Line 349-350, word choice of "high concentration of enterprises", "agglomeration" would be better.

Line 389, word choice of "heaviest", "highest" would be better.

Line 398-399, word choice of "ultra-low transformation", "lowest" would be better.

Line 498/ 521/ 533, what are the meaning of 10%, 5%, and 1%? How about the p-value?

Line 537, word choice of "metropolis", "city" would be better.

Moreover, it would be better to combine 1. Introduction and 2. Literature review and makes the article look concise.

7. PLOS authors have the option to publish the peer review history of their article (what does this mean?). If published, this will include your full peer review and any attached files.

Reviewer #1: No

Reviewer #2: No

---

## [Author Response · Author response to Decision Letter 1]

1 Jun 2021

Response to the Editors and reviewers

1. Responses to the Editors:

Comment (1): 

If applicable, we recommend that you deposit your laboratory protocols in protocols.io to enhance the reproducibility of your results. Protocols.io assigns your protocol its own identifier (DOI) so that it can be cited independently in the future. For instructions see: http://journals.plos.org/plosone/s/submission-guidelines#loc-laboratory-protocols.

Response: Thank you for your kind advice. We will deposit this protocol in future. We have also modified and improved the manuscript according to the reviewers’ comments, with a particular focus on improving language.

Comment (2): 

Response: 

Thank you for your suggestion. We have reviewed the reference list and ensured that we have not cited papers that have been retracted.

Comment (3):

Is the manuscript technically sound, and do the data support the conclusions? The manuscript must describe a technically sound piece of scientific research with data that supports the conclusions. Experiments must have been conducted rigorously, with appropriate controls, replication, and sample sizes. The conclusions must be drawn appropriately based on the data presented.

Reviewer #1: Yes

Reviewer #2: Partly 

Response to Reviewer #2:

We are very much appreciative of the time you have taken to review our revised manuscript and provide valuable comments. The relationship among enterprises’ agglomeration, pollution emissions, and air quality is complex; as such, there may be factors that cannot be reflected in the data. There are various methods to measure agglomeration; we believe that EGI and HHI are appropriate and have provided rationales to this end. The data were examined and the data quality was guaranteed, and the sample size was sufficient. From our limited understanding, we believe the manuscript is technically sound, and the data can support the conclusions. The manuscript also provides some novel examples and perspectives that we would really like to share with other researchers.

Comment (4): 

Is the manuscript presented in an intelligible fashion and written in standard English?

Reviewer #1: Yes

Reviewer #2: No 

Response to Reviewer #2: 

Thank you for your kind advice and comments. We appreciate the time you have put into reviewing the manuscript. The manuscript has been edited by a Language Editing Service to ensure that the submitted article is clear, correct, and unambiguous to readers. We have uploaded the language editing certificate to the attachments.

Comment (5): 

The modified version is clearer than the original, but proofread might be helpful to improve scientific writing. Please check all the article again. For example,

Line1-13, Font inconsistency with other parts of article.

Line 42, Line 88, Word choice of "externalities", "impact" would be better.

Line 45, generates more "pollutants", "pollution" would be better.

Line 47, "production concentration", The usage seems weird, and would be better to change it.

Line 48, 545, word choice of "spillover effect", Would it be better to replace a word or not used so many times.

Line 51-54, 399, “Beijing-Tianjin- Hebei (BTH) Collaborative Development Plan", “Opinions on Strengthening the Construction of the Key Platform for the Transfer of the Beijing-Tianjin-Hebei (BTH) Industry”, "Blue-Sky Protection Campaign" would be better written in italics.

Line 63, emission intensity per unit of land area, "per square kilometers" would be better.

Line 82-83, word choice of "agglomeration", "pollution aggregation" would be better.

Line 103, word choice of "pollutant emission", "pollution" would be better.

Line 114, Some Literature, "L" should not use capitalized.

Line 119, Some literature shows, It should be plural and no "s".

Line 239, Spatial correlation analysis versus Line 438, Spatial autocorrelation test, Uniform usage is better.

Line 349-350, word choice of "high concentration of enterprises", "agglomeration" would be better.

Line 389, word choice of "heaviest", "highest" would be better.

Line 398-399, word choice of "ultra-low transformation", "lowest" would be better.

Line 498/ 521/ 533, what are the meaning of 10%, 5%, and 1%? How about the p-value?

Line 537, word choice of "metropolis", "city" would be better.

Response: 

Thank you for reviewing the language issues in this manuscript. We have checked the manuscript again, and have revised much of the article, whereby details of this article have been improved. The revised manuscript has been edited by a Language Editing Service to ensure that the submitted article is clear, correct, and unambiguous to readers. We have uploaded the language editing certificate in the attachments. Thank you again for your careful proofreading and comments.

Comment (6): 

it would be better to combine 1. Introduction and 2. Literature review and makes the article look concise.

Response: 

Thank you for your kind advice and comments. We have revised the literature review and Introduction sections and combined them to provide a more concise introduction.

---

## [Decision Letter · Decision Letter 2]

9 Jul 2021

Industrial Agglomeration and Air Pollution: A New Perspective from Enterprises in Atmospheric Pollution Transmission Channel Cities (APTCC) of Beijing-Tianjin-Hebei and Surrounding Areas, China

PONE-D-20-38132R2

Dear Dr. Zhou,

We’re pleased to inform you that your manuscript has been judged scientifically suitable for publication and will be formally accepted for publication once it meets all outstanding technical requirements.

Kind regards,

Chon-Lin Lee, Ph.D.

Academic Editor

PLOS ONE

Additional Editor Comments (optional):

Reviewers' comments:

Reviewer's Responses to Questions

**Comments to the Author**

1. If the authors have adequately addressed your comments raised in a previous round of review and you feel that this manuscript is now acceptable for publication, you may indicate that here to bypass the “Comments to the Author” section, enter your conflict of interest statement in the “Confidential to Editor” section, and submit your "Accept" recommendation.

Reviewer #2: All comments have been addressed

2. Is the manuscript technically sound, and do the data support the conclusions?

Reviewer #2: Yes

3. Has the statistical analysis been performed appropriately and rigorously? 

Reviewer #2: Yes

4. Have the authors made all data underlying the findings in their manuscript fully available?

Reviewer #2: Yes

5. Is the manuscript presented in an intelligible fashion and written in standard English?

Reviewer #2: Yes

6. Review Comments to the Author

Reviewer #2: Basically, this revision has almost corrected the comments made last time, and there are no more comments this time.

7. PLOS authors have the option to publish the peer review history of their article (what does this mean?). If published, this will include your full peer review and any attached files.

Reviewer #2: No

---

## [Editor Report · Acceptance letter]

14 Jul 2021

PONE-D-20-38132R2 

Industrial Agglomeration and Air Pollution: A New Perspective from Enterprises in Atmospheric Pollution Transmission Channel Cities (APTCC) of Beijing-Tianjin-Hebei and its Surrounding Areas, China 

Dear Dr. Zhou:

I'm pleased to inform you that your manuscript has been deemed suitable for publication in PLOS ONE. Congratulations! Your manuscript is now with our production department. 

Kind regards, 

on behalf of

Dr. Chon-Lin Lee 

Academic Editor

PLOS ONE